# dgMARK: Decoding-Guided Watermarking for Diffusion Language Models

**Pyo Min Hong** [1]   **Albert No** [2]

## Abstract

We propose **dgMARK**, a decoding-guided watermarking method for discrete diffusion language models (dLLMs). Unlike autoregressive models, dLLMs can generate tokens in arbitrary order. While an ideal conditional predictor would be invariant to this order, practical dLLMs exhibit strong sensitivity to the unmasking order, creating a new channel for watermarking. dgMARK steers the unmasking order toward positions whose high-reward candidate tokens satisfy a simple parity constraint induced by a binary hash, *without explicitly reweighting* the model's learned probabilities. The method is plug-and-play with common decoding strategies (e.g., confidence, entropy, and margin-based ordering) and can be strengthened with a one-step lookahead variant. Watermarks are detected via elevated parity-matching statistics, and a sliding-window detector ensures robustness under post-editing operations including insertion, deletion, substitution, and paraphrasing. Project website: https://dgmark-watermarking.github.io

## 1. Introduction

Large Language Models (LLMs) now generate coherent and high-quality text, enabling applications in question answering (Yue, 2025), programming (Jiang et al., 2026a), and academic writing (Perkins, 2023). At the same time, this capability raises serious risks: machine-generated content can be weaponized for disinformation (Ranade et al., 2021), phishing (Karanjai, 2022), and plagiarism (Kasneci et al., 2023), and may exacerbate copyright infringement (Rillig et al., 2023), identity theft (Kumar et al., 2024), and fraud (Mirsky et al., 2023). As LLMs become more accessible, reliable provenance tools that distinguish machine-generated from human-authored text are increasingly important (Bender et al., 2021; Crothers et al., 2023).

Watermarking is one of the most practical approaches to content provenance. LLM watermarking methods embed subtle statistical signals that can later be detected (Feng et al., 2025; Wu et al., 2025a). Most existing schemes, however, are designed for autoregressive models (ARMs) and assume left-to-right generation. A prominent class explicitly *biases* token probabilities, typically by partitioning the vocabulary into "green" and "red" lists and shifting mass toward green tokens (Kirchenbauer et al., 2023), which can degrade text quality. Another line aims to reduce distortion by conditioning sampling on long pseudo-random keys (Kuditipudi et al., 2024), but may incur slower detection and limited scalability. Crucially, both paradigms rely on a fixed causal context (i.e., previous tokens or $n$-grams), which is unavailable when generation is not strictly left-to-right.

Recently, *discrete diffusion language models* (dLLMs) (Lou et al., 2024; Nie et al., 2025) have emerged as a strong alternative to the autoregressive paradigm. dLLMs iteratively denoise masked sequences and can finalize tokens in *arbitrary order*, supporting adaptive decoding strategies and controllable generation (Yu et al., 2025; Li et al., 2025). This order-agnostic decoding both creates challenges and opens new opportunities for watermarking. On one hand, the "previous token" used by ARM watermarks is often undefined during diffusion decoding. On the other hand, dLLMs expose the *decoding order* as a new control knob. In principle, if a dLLM learned all conditional distributions perfectly, the unmasking order would not affect the resulting generation statistics. In practice, however, decoding order *does* matter: recent analysis shows that choosing the decoding order adaptively can substantially change generation quality and behavior in masked diffusion models (Kim et al., 2025). This gap between ideal order-invariance and practical order-sensitivity suggests an appealing direction: embed watermarks by steering the decoding order, *without explicitly reweighting the learned probabilities*.

Very recent preprints have begun exploring watermarking for dLLMs (Bagchi et al., 2025; Wu et al., 2025b; Gloaguen et al., 2026; Raban et al., 2026). These works address the missing left-to-right context via predictive or bidirectional context construction, or via controlled sampling procedures

---

[1]Department of Computer Engineering, Hongik University
[2]Department of Artificial Intelligence, Yonsei University. Correspondence to: Albert No <albertno@yonsei.ac.kr>.

*Proceedings of the 43$^{rd}$ International Conference on Machine Learning*, Seoul, South Korea. PMLR 306, 2026. Copyright 2026 by the author(s).

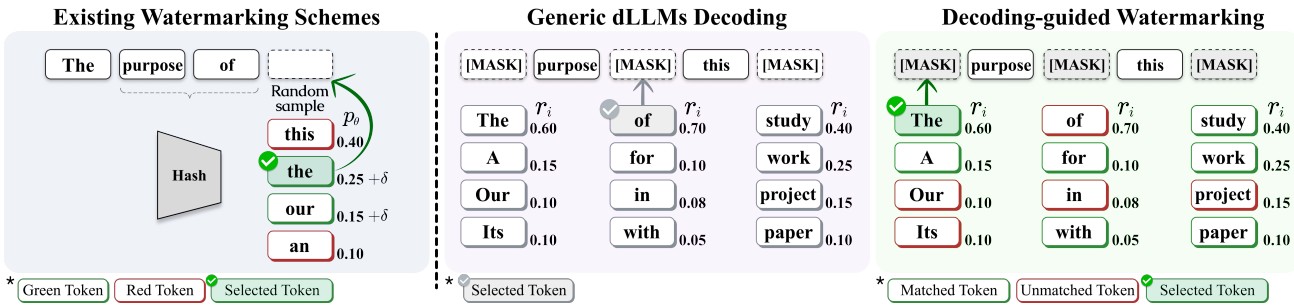

*Figure 1.* **Overview. (Left)** Existing autoregressive watermarking methods generate green/red token sets by hashing the preceding context and embed watermark signals by biasing the sampling distribution toward green tokens. **(Middle)** In contrast, decoding in dLLMs does not follow the traditional left-to-right generation process; instead, the model selects high-reward tokens at each position even in the absence of prior context. **(Right)** The proposed method leverages these rewards and embeds watermark signals by prioritizing tokens with high reward that satisfy the parity condition.

(e.g., seeded randomness) to enable detection. While effective, most approaches still embed signals by altering the token selection probability, and do not exploit decoding order itself as the primary watermarking channel.

This paper introduces **dgMARK**, a decoding-guided watermarking method for dLLMs (Figure 1). Unlike probability-biasing watermarks, dgMARK does *not* manually alter or reweight the model's learned probabilities. Instead, it embeds a watermark by guiding *which position is unmasked next*. At each step, we apply a lightweight binary hash rule and prioritize decoding positions whose high-reward candidates satisfy a parity constraint tied to the position index. Over a full sequence, this induces a systematic increase in the parity-matching rate above the random baseline (0.5), which serves as the test statistic for detection. Because the method operates as a wrapper around the decoding policy, it is compatible with common dLLM decoding strategies (e.g., confidence-, entropy-, or margin-based ordering), and it can be strengthened by a one-step lookahead variant. To ensure robustness to post-editing (insertion, deletion, and substitution), we use a sliding-window detector that captures characteristic local deviations caused by alignment shifts.

We evaluate dgMARK on benchmark datasets using state-of-the-art dLLMs, including LLaDA and Dream. Across models and tasks, dgMARK achieves strong detectability with minimal degradation in text quality, and remains robust under extensive post-editing and paraphrasing. These results demonstrate that decoding order provides a practical and complementary watermarking channel for dLLMs.

## 2. Related Work

**Discrete diffusion language models (dLLMs).** Diffusion models (Ho et al., 2020; Song et al., 2021b;a) have achieved strong results in continuous domains such as images (Rombach et al., 2022; Saharia et al., 2022) and have been adapted to discrete domains through Masked Diffusion Models (MDMs) (Austin et al., 2021; Lou et al., 2024; Sa-

hoo et al., 2024; Shi et al., 2024; Ou et al., 2025), which iteratively denoise masked tokens. A key property of MDMs and dLLMs is *order-agnostic generation*: they model conditional distributions under arbitrary masking patterns, admitting a wide range of decoding strategies (e.g., random, confidence-, entropy-, and margin-based ordering). Recent large-scale dLLMs such as LLaDA (Nie et al., 2025; Zhu et al., 2025; Bie et al., 2025) and Dream (Ye et al., 2025) demonstrate that this paradigm scales competitively, often matching or surpassing autoregressive models in data-constrained regimes (Prabhudesai et al., 2026). Industrial systems including Mercury (Labs et al., 2025) and Gemini Diffusion (DeepMind, 2025) further highlight the practical efficiency of diffusion-based decoding. Finally, recent analyses show that while ideal models would be order-invariant, practical dLLMs can be sensitive to the decoding order, motivating algorithmic designs that leverage decoding strategies as a first-class control knob (Kim et al., 2025).

**Watermarking in LLMs.** Digital watermarking has long been used to trace provenance and embed imperceptible signals across text, images, and other media (Petitcolas et al., 1999; Zhu et al., 2018; Liang et al., 2026). In LLMs, a dominant line of work embeds watermarks by *biasing token probabilities* during generation. Representative methods (Kirchenbauer et al., 2023; Zhao et al., 2023; 2024) partition the vocabulary into "green" and "red" sets and increase the probability of sampling green tokens; detection is then performed via statistical tests on the frequency of green tokens. Although these methods can provide strong detectability guarantees, probability biasing alters the model's output distribution and may degrade text quality. To mitigate this, distortion-free variants (Kuditipudi et al., 2024; Christ et al., 2024) aim to preserve the original distribution, often by conditioning sampling on a long pseudorandom sequence derived from a secret key, which may introduce additional computation compared to simple probability biasing. Aaronson & Kirchner (2023) adopt the GumbelMax-trick to ensure distortion-free generation, using an exponential repa-

rameterization of sampling, and subsequent studies extend this approach (Fu et al., 2024). Despite their differences, both probability biasing and distortion-free approaches are fundamentally tailored to left-to-right generation.

**Watermarking beyond left-to-right generation.** Recent work has explored watermarking for settings where generation is not strictly autoregressive. For example, Chen et al. (2025) propose watermarking frameworks for order-agnostic models, but still rely on biasing the token selection mechanism, thereby retaining a detectability–quality trade-off. Very recent concurrent preprints have begun studying watermarking specifically for diffusion-style language models and dLLMs (Bagchi et al., 2025; Wu et al., 2025b; Gloaguen et al., 2026; Raban et al., 2026). While these works differ in their embedding and detection choices, a common theme is to recover watermark signals by directly controlling token selection (e.g., via probability shaping, controlled sampling, or key-conditioned token constraints) rather than exploiting decoding-order degrees of freedom.

## 3. Decoding-guided Watermarking for dLLMs

### 3.1. Generic Decoding Strategy

Discrete diffusion language models (dLLMs) differ from autoregressive models in a crucial way: instead of being forced to generate text strictly from left to right, they can in principle reveal tokens in *any order*. This property arises because dLLMs are trained to predict a missing token given an arbitrary subset of revealed tokens. As a result, the same sequence can be generated through many different decoding orders, making decoding strategy an essential design choice.

Formally, let $p_{\text{data}}$ denote the true data distribution. Given a prompt $x = (x_1, \ldots, x_m)$, the goal of dLLMs is to generate a sequence $y = (y_1, \ldots, y_n)$ such that $y \sim p_{\text{data}}(y|x)$. For any subset of revealed indices $\mathcal{I} \subset \{1, \ldots, n\}$ (where $y_j$ is revealed for $j \in \mathcal{I}$) and a target index $i \notin \mathcal{I}$, the dLLMs learn a predictor $p_\theta$ that approximates

$$p_\theta(y_i|y_{\mathcal{I}}, x) \approx p_{\text{data}}(y_i|y_{\mathcal{I}}, x),$$

while treating the remaining tokens as [MASK]. This means that, ideally, the distribution of $y$ can be factorized along any permutation $\pi$ of $\{1, \ldots, n\}$:

$$p_{\text{data}}(y|x) = \prod_{i=1}^{n} p_{\text{data}}(y_{\pi(i)}|y_{\pi(<i)}, x)$$
$$\approx \prod_{i=1}^{n} p_\theta(y_{\pi(i)}|y_{\pi(<i)}, x),$$

where $y_{\pi(<i)} = \{y_{\pi(k)}|k < i\}$. In theory, the choice of order $\pi$ should not matter. In practice, however, imperfect training causes different decoding strategies to yield

different results, making the decoding strategy a central component of dLLMs generation (Kim et al., 2025). Accordingly, the watermark signal observed in practice arises from approximation error and decoding heuristics.

At each decoding step $i$, let $\mathcal{I} = \{\pi(1), \ldots, \pi(i-1)\}$ be the set of revealed indices. A decoding strategy $\mathcal{F}(j; p_\theta, x, y_{\mathcal{I}})$ returns, for each unrevealed index $j \notin \mathcal{I}$, a reward $r_j$ and a sampled candidate token $v_j$. The next index is then chosen as $\pi(i) = \arg\max_{j \notin \mathcal{I}} r_j$, and the corresponding token will be $y_{\pi(i)} \leftarrow v_{\pi(i)}$. This generic decoding procedure is summarized in Algorithm 1.

A range of decoding strategies $\mathcal{F}$ have been proposed, reflecting trade-offs between certainty and exploration (Nie et al., 2025; Ye et al., 2025; Kim et al., 2025). Common examples include:

- **Random**: rewards $r_j$ are sampled uniformly at random; sample $v_j \sim p_\theta(y_j = \cdot|y_{\mathcal{I}}, x)$.

- **Confidence**: sample $v_j \sim p_\theta(y_j = \cdot|y_{\mathcal{I}}, x)$ and set $r_j = p_\theta(y_j = v_j|y_{\mathcal{I}}, x)$.

- **Entropy**: set $r_j = -H(Y_j|y_{\mathcal{I}}, x)$, where $H(\cdot)$ denotes the conditional entropy under $p_\theta$.

- **Margin**: let $v_j = \arg\max_v p_\theta(v|y_{\mathcal{I}}, x)$ and define $r_j$ as the probability gap between the top-1 and top-2 candidates.

In decoding strategies that involve stochastic sampling, the token $v_j$ may either be drawn from $p_\theta(\cdot|y_{\mathcal{I}}, x)$ or chosen greedily as $v_j = \arg\max_v p_\theta(y_j = v|y_{\mathcal{I}}, x)$.

Finally, although parallel decoding methods exist that reveal multiple tokens at once to speed up generation (Ben-Hamu et al., 2025; Wei et al., 2026), here we focus on the sequential framework.

### 3.2. Problem Setup

Our goal is to design a watermarking strategy for discrete diffusion language models (dLLMs). In this setting, watermarking means that a sequence $y \sim p_{\text{data}}(y|x)$ and a sequence $y' \sim p_\theta(y|x)$ generated by a designated dLLM can be made *statistically distinguishable* given a secret key, while maintaining high text quality.

Most existing watermarking methods for LLMs achieve detectability by *explicitly biasing* token probabilities. Although effective, such probability shaping changes the model's sampling distribution and can introduce a direct detectability–quality trade-off. In contrast, dLLMs provide a different handle: since they generate by iteratively unmasking tokens and can, in principle, decode in arbitrary orders, watermarking can be implemented by modifying the

**Algorithm 1** Generic dLLMs Decoding

**Require:** Prompt $x$; output length $n$; predictor $p_\theta$; decoding strategy $\mathcal{F}$
1: $y \leftarrow [\text{MASK}]^n$;    $\mathcal{I} \leftarrow \emptyset$
2: **for** $i = 1, \ldots, n$ **do**
3:     Get $\{(r_j, v_j) = \mathcal{F}(j; p_\theta, x, y_\mathcal{I}) \mid j \notin \mathcal{I}\}$
4:     $\mathcal{C} \leftarrow \{j \notin \mathcal{I}\}$
5:     $k^\star \leftarrow \arg\max_{j \in \mathcal{C}} r_j$
6:     $y_{k^\star} \leftarrow v_{k^\star}$;       $\mathcal{I} \leftarrow \mathcal{I} \cup \{k^\star\}$
7: **end for**
8: **Return** $y$

**Algorithm 2** dgMARK: Watermarks by Decoding

**Require:** Prompt $x$; output length $n$; predictor $p_\theta$; decoding strategy $\mathcal{F}$; matching set $\mathcal{G}_j$
1: $y \leftarrow [\text{MASK}]^n$;    $\mathcal{I} \leftarrow \emptyset$
2: **for** $i = 1, \ldots, n$ **do**
3:     Get $\{(r_j, v_j) = \mathcal{F}(j; p_\theta, x, y_\mathcal{I}) \mid j \notin \mathcal{I}\}$
4:     $\mathcal{C} \leftarrow \{j \notin \mathcal{I} \mid v_j \in \mathcal{G}_j\}$
5:     **if** $\mathcal{C} = \emptyset$ **then** $\mathcal{C} \leftarrow \{j \notin \mathcal{I}\}$ **end if**
6:     $k^\star \leftarrow \arg\max_{j \in \mathcal{C}} r_j$
7:     $y_{k^\star} \leftarrow v_{k^\star}$;       $\mathcal{I} \leftarrow \mathcal{I} \cup \{k^\star\}$
8: **end for**
9: **Return** $y$

*decoding strategy* (i.e., the unmasking order) rather than manually reweighting token probabilities.

Accordingly, we focus on watermarking that embed signals through the decoding process itself. Concretely, we design an *adaptive ordering strategy* that decides which position to unmask next in a way that enhances detectability, while using the model's conditional predictor $p_\theta(y_j | y_\mathcal{I}, x)$ without direct logit or probability modification (i.e., without explicit probability reweighting). In our experiments, this design preserves text quality to a large extent while enabling reliable watermark detection.

We formalize watermarking for dLLMs as a decoding problem with the following requirements:

1. **No direct probability reweighting**: The watermarking procedure should not directly modify the model's learned conditional probabilities, for example through logit biasing, green/red probability boosting, or distribution renormalization.

2. **Detectability**: The watermark must be verifiable from the generated text $y'$ and the secret key alone, without requiring access to the model internals or the prompt.

3. **Robustness**: The watermark must remain detectable under random or adversarial post-editing (e.g., insertions, deletions, and substitutions).

### 3.3. dgMARK: Decoding-guided Watermarking

We now present **dgMARK**, our watermarking method for dLLMs. The approach is inspired by prior LLM watermarking schemes (Kirchenbauer et al., 2023), which conceptually divide the vocabulary into two groups and analyze the frequency of designated tokens. In contrast to those methods, which modify token probabilities, dgMARK embeds the watermark by guiding the decoding order. A lightweight binary hashing rule determines which candidate tokens align with the position index, and the decoder simply prioritizes those positions. This embeds a detectable signal while leaving the learned probabilities $p_\theta(y_j | y_\mathcal{I}, x)$ unchanged.

Given a watermark key $\xi$, we define a deterministic hashing function $f : \mathcal{V} \times \Xi \to \{0, 1\}$ that maps each token $v \in \mathcal{V}$ to a binary value conditioned on $\xi$. The function is constructed so that, for any key $\xi$, the resulting partition is balanced. At each position $i$, the vocabulary is divided as

$$\mathcal{G}_i = \{v \in \mathcal{V} \mid f(v, \xi) \equiv i \pmod 2\}, \qquad \mathcal{R}_i = \mathcal{V} \setminus \mathcal{G}_i,$$

where $\mathcal{G}_i$ is the parity-matching set and $\mathcal{R}_i$ is the residual.

During decoding, we prioritize indices $j$ whose predicted tokens satisfy the parity condition (i.e., $v_j \in \mathcal{G}_j$); among these, we select the index with the largest reward. If no such index exists, the procedure falls back to $\mathcal{R}_i$. This strategy is summarized in Algorithm 2.

The dgMARK framework is compatible with any decoding strategy $\mathcal{F}$. Within dgMARK, the decoding order $\pi$ is adjusted so that positions with parity-matching candidates are filled first, while remaining positions are handled afterward. The generated text remains visually indistinguishable from standard decoding, while its parity-matching rate is systematically higher than chance, providing a reliable statistical watermark signal.

### 3.4. dgMARK with One-step Lookahead Beam Search

The standard version of dgMARK selects the next index $k^\star$ with the highest reward among *parity-matching* candidates, i.e., indices $j \notin \mathcal{I}$ such that $v_j \in \mathcal{G}_j$. Although straightforward, this greedy choice can commit too early to a locally optimal decision, which may reduce the number of parity-matching opportunities available in subsequent steps.

To mitigate this issue, we introduce a top-$k$ lookahead variant. At each step, we first form the candidate set

$$\mathcal{C} = \{j \notin \mathcal{I} \mid v_j \in \mathcal{G}_j\},$$

and if $\mathcal{C} = \emptyset$ we fall back to $\mathcal{C} = \{j \notin \mathcal{I}\}$ as in standard dgMARK. We then select the top-$k$ indices $\mathcal{T} \subseteq \mathcal{C}$ with the largest rewards $r_j$.

For each candidate $j \in \mathcal{T}$, we compute a one-step lookahead

score that estimates how many *next-step* candidates remain parity-matching after committing to $y_j \leftarrow v_j$. Concretely, let $(r_\ell^{(j)}, v_\ell^{(j)})$ denote the outputs of the same decoding strategy $\mathcal{F}$ applied to the updated partial sequence after setting $y_j \leftarrow v_j$ (with revealed set $\mathcal{I} \cup \{j\}$). We define

$$g^{(j)} = \sum_{\ell \notin \mathcal{I} \cup \{j\}} \mathbb{1}\left[ v_\ell^{(j)} \in \mathcal{G}_\ell \right].$$

Finally, we select

$$k^\star = \arg\max_{j \in \mathcal{T}} g^{(j)},$$

with ties broken by $r_j$.

When $k = 1$, this reduces to the standard dgMARK (greedy strategy). Larger $k$ trades additional computation for stronger watermark embedding by explicitly preserving future parity-alignment opportunities.

### 3.5. Watermark Detection

Given a generated sequence $y = (y_1, \ldots, y_n)$ and the secret key $\xi$ (used by the hash function $f$), we detect the watermark by measuring how often tokens satisfy the parity condition. Equivalently, define the parity-matching set $\mathcal{G}_i = \{v \in \mathcal{V} \mid f(v, \xi) \equiv i \pmod 2\}$ and check whether $y_i \in \mathcal{G}_i$.

**Basic Detection (global $z$-test).** Define

$$m_i = \mathbb{1}[y_i \in \mathcal{G}_i], \qquad G = \sum_{i=1}^{n} m_i.$$

Because $f(\cdot, \xi)$ is balanced, under non-watermarked generation we have $\mathbb{E}[m_i] \approx \frac{1}{2}$ and $G \approx \mathrm{Binomial}(n, \frac{1}{2})$. We compute the standard $z$-score

$$z = \frac{G - n/2}{\sqrt{n/4}}. \tag{1}$$

We declare $y$ as watermarked if $z$ exceeds a chosen threshold (one-sided test).

**Robust Detection (sliding-window $z$-statistics).** To handle post-editing (e.g., insertion, deletion, and substitution), we apply the same test on overlapping windows. For a window length $w$ and start index $s$, define

$$G_s = \sum_{i=s}^{s+w-1} \mathbb{1}[y_i \in \mathcal{G}_i], \qquad z_s = \frac{G_s - w/2}{\sqrt{w/4}}.$$

Insertions or deletions may shift indices and invert parity alignment, causing some windows to deviate *below* $\frac{1}{2}$. We therefore use a two-sided window aggregated statistic, e.g.,

$$z_{\text{win}} = \frac{1}{S} \sum_{s=1}^{S} z_s^2,$$

and declare the sequence watermarked if $z_{\text{win}}$ exceeds a threshold.

*Table 1.* **Empirical error rates.** The results are based on texts generated by multiple dLLMs with multinomial sampling.

| Dataset | Model | PPL | $z = 4.0$ | | | |
|---|---|---|---|---|---|---|
| | | | FPR↓ | TNR↑ | TPR↑ | FNR↓ |
| C4 | LLaDA | 4.90 | 0.000 | 1.000 | 0.957 | 0.043 |
| | LLaDA 1.5 | 5.27 | 0.000 | 1.000 | 0.929 | 0.071 |
| | Dream | 5.75 | 0.000 | 1.000 | 0.958 | 0.042 |
| Writing Prompts | LLaDA | 6.00 | 0.000 | 1.000 | 0.983 | 0.017 |
| | LLaDA 1.5 | 6.34 | 0.004 | 0.996 | 0.733 | 0.267 |
| | Dream | 6.87 | 0.007 | 0.993 | 0.682 | 0.318 |

## 4. Experimental Evaluation

### 4.1. Experimental Setup

**Datasets and Prompts.** We use two benchmark datasets. The first is the news-like subset of C4 (Raffel et al., 2023), which has been widely employed in prior watermarking studies (Kirchenbauer et al., 2023; Kuditipudi et al., 2024; Block et al., 2025; Feng et al., 2025). The second is Writing Prompts (Fan et al., 2018), which provides diverse topics and narrative styles, ranging from apocalyptic scenarios to everyday stories. For C4, we randomly sample texts and truncate them to a fixed length to serve as prompts; for Writing Prompts, the given prompts are used directly.

**Models and Environments.** Experiments are conducted on LLaDA-8B (Nie et al., 2025), LLaDA 1.5-8B (Zhu et al., 2025), LLaDA 2.0-mini-16B (Bie et al., 2025), and Dream-7B (Ye et al., 2025). All models used in our experiments are instruction-following models and generate sequences of length 256 using block-wise generation (Arriola et al., 2025; Nie et al., 2025). We adopt block sizes of 32 for the LLaDA family and 8 for Dream-7B to encourage longer outputs. Responses are generated for 300 prompts, and we retain sequences longer than 200 tokens (100 tokens for Dream-7B, due to its shorter generations). Text quality is evaluated using perplexity (PPL) computed with Gemma3-12B (Team et al., 2025), a larger model serving as an oracle. For LLaDA 2.0, we use a quantized LLaMA 3.3-70B, which preserves relative trends across watermarking methods.

**Sampling Schemes.** We consider two sampling schemes: multinomial sampling, where tokens are drawn from $p_\theta$, and greedy sampling, where the most likely token is chosen at each step. In both settings, the decoding strategy $\mathcal{F}$ follows the confidence rule unless otherwise stated. Beam search, as described in Section 3.4, augments these schemes with one-step lookahead. Additional experiments with entropy and margin-based decoding are reported in the Appendix.

**Evaluation Metrics.** Performance is assessed along three axes: (1) Detectability: measured by $z$-score (Eq. (1)), false positive rate (FPR), false negative rate (FNR), and true posi-

*Table 2.* **Watermark Detectability Comparison.** Empirical results under greedy and multinomial sampling with LLaDA 1.5 on the C4 dataset, reporting perplexity (PPL) and detection metrics. Greedy and multinomial sampling represent the non-watermarked baselines.

| Method | PPL ↓ | FPR↓ | TNR↑ | TPR↑ | FNR↓ | TPR@FPR ↑ | | | |
| --- | --- | --- | --- | --- | --- | --- | --- | --- | --- |
| | | | | | | 10% | 1% | 0.1% | 0.01% |
| **Greedy Sampling** | 4.03 | - | - | - | - | - | - | - | - |
| KGW ($\delta = 1$) | 4.33 | 0.0 | 1.0 | 0.072 | 0.928 | 88.52 | 62.68 | 30.14 | 11.48 |
| KGW ($\delta = 2$) | 5.02 | 0.0 | 1.0 | 0.866 | 0.134 | 100.00 | 97.31 | 93.01 | 97.63 |
| KGW ($\delta = 3$) | 5.83 | 0.0 | 1.0 | 0.970 | 0.030 | 100.00 | 100.00 | 98.52 | 97.78 |
| PATTERN-MARK ($\delta = 1$) | 4.11 | 0.0 | 1.0 | 0.000 | 1.000 | 21.76 | 4.17 | 1.39 | 0.00 |
| PATTERN-MARK ($\delta = 2$) | 4.72 | 0.0 | 1.0 | 0.040 | 0.960 | 73.50 | 48.50 | 20.50 | 12.00 |
| PATTERN-MARK ($\delta = 3$) | 5.86 | 0.0 | 1.0 | 0.584 | 0.416 | 96.26 | 91.59 | 87.38 | 78.97 |
| dgMARK | **4.44** | 0.0 | 1.0 | 0.540 | 0.460 | 97.86 | 91.98 | 76.47 | 60.96 |
| + 3-beam | 4.75 | 0.0 | 1.0 | 0.963 | 0.037 | 100.00 | 99.54 | 98.62 | 97.25 |
| + 5-beam | 5.01 | 0.0 | 1.0 | 0.987 | 0.013 | 100.00 | 100.00 | 99.56 | 98.69 |
| + 8-beam | 5.16 | 0.0 | 1.0 | 0.991 | 0.008 | 100.00 | 100.00 | 100.00 | 99.12 |
| **Multinomial Sampling** | 4.21 | - | - | - | - | - | - | - | - |
| KGW ($\delta = 1$) | 5.59 | 0.0 | 1.0 | 0.107 | 0.893 | 89.80 | 60.91 | 32.99 | 14.21 |
| KGW ($\delta = 2$) | 6.38 | 0.0 | 1.0 | 0.876 | 0.124 | 99.41 | 98.82 | 97.65 | 91.18 |
| KGW ($\delta = 3$) | 7.87 | 0.0 | 1.0 | 0.984 | 0.016 | 100.00 | 99.21 | 99.21 | 98.41 |
| PATTERN-MARK ($\delta = 1$) | 5.45 | 0.0 | 1.0 | 0.000 | 1.000 | 25.26 | 5.67 | 1.55 | 0.00 |
| PATTERN-MARK ($\delta = 2$) | 6.33 | 0.0 | 1.0 | 0.060 | 0.940 | 78.00 | 53.50 | 27.50 | 16.50 |
| PATTERN-MARK ($\delta = 3$) | 7.69 | 0.0 | 1.0 | 0.586 | 0.414 | 98.99 | 95.96 | 91.41 | 83.33 |
| dgMARK | **5.27** | 0.0 | 1.0 | 0.929 | 0.071 | 100.00 | 100.00 | 99.41 | 95.29 |
| + 3-beam | 5.40 | 0.0 | 1.0 | 1.000 | 0.000 | 100.00 | 100.00 | 100.00 | 100.00 |
| + 5-beam | 5.76 | 0.0 | 1.0 | 1.000 | 0.000 | 100.00 | 100.00 | 100.00 | 100.00 |
| + 8-beam | 6.00 | 0.0 | 1.0 | 1.000 | 0.000 | 100.00 | 100.00 | 100.00 | 100.00 |

tive rate at a fixed false positive rate (TPR@FPR). (2) Text Quality: measured by perplexity (PPL) and benchmark accuracy. We include three representative benchmarks using the lm-evaluation-harness (Gao et al., 2023) to measure downstream capability: MMLU (Hendrycks et al., 2021) (multi-task reasoning), GSM8K (Cobbe et al., 2021) (mathematical problem solving), and HumanEval (Chen et al., 2021) (code generation). (3) Robustness: measured by ROC curves under token-level editing and paraphrasing attacks.

**Baselines.** For watermarking baselines, we compare against two existing methods: KGW (Kirchenbauer et al., 2023), a representative autoregressive watermarking method, and PATTERN-MARK (Chen et al., 2025), a watermarking method designed for order-agnostic models. As naive application of KGW to masked tokens results in unreliable detection due to violated prefix-based assumptions (Wu et al., 2025b), results are derived from dLLMs configured to generate tokens sequentially from left to right.

### 4.2. Experimental Analyses

**Watermark Detectability.** Table 1 summarizes the performance of standard dgMARK across models and datasets, confirming reliable detection with negligible error rates. Table 2 compares dgMARK against two baselines, KGW and PATTERN-MARK, under multiple watermark strengths ($\delta \in \{1, 2, 3\}$). Overall, dgMARK with 3-beam search

achieves the strongest detectability, while incurring a smaller increase in perplexity than the baselines. Additional results on LLaDA 2.0 and illustrative generations are provided in the Appendix, showing a higher parity-matching ratio for watermarked text than for non-watermarked text.

**Effect of Sampling Schemes.** Table 2 compares greedy sampling and multinomial sampling with and without beam search. As beam size increases ($k \in \{1, 3, 5, 8\}$), error rates consistently decrease, demonstrating that one-step lookahead strengthens parity alignment. Multinomial sampling generally yields higher perplexity but produces stronger watermark signals than greedy sampling.

**Direct vs. Implicit Effects on the Output Distribution.** Our claim is not that dgMARK preserves the final output distribution. Rather, dgMARK does not *directly* modify the model's learned conditional probabilities through logit biasing, renormalization, or explicit green/red boosting. In the greedy setting, the standard dLLM decoding procedure already converts each conditional distribution into a proposal $v_j = \arg\max_v p_\theta(v|y_\mathcal{I}, x)$. dgMARK follows the same convention and changes only which position-token proposal is committed next, preferring proposals whose current token satisfies the parity condition. In the multinomial setting, the dLLM implementation used in our experiments follows a sample-then-select procedure: candidate tokens

*Table 3.* **Benchmark results.** Evaluated on LLaDA, LLaDA 1.5, and Dream under greedy and multinomial sampling. The comparison includes (1) non-watermarked baseline, (2) KGW, (3) PATTERN-MARK, (4) dgMARK, and (5) dgMARK with 3-beam search.

| Model | Method | Greedy Sampling | | | Multinomial Sampling | | |
|---|---|---|---|---|---|---|---|
| | | MMLU (Acc ↑) | GSM8K (Acc ↑) | HumanEval (Pass@1 ↑) | MMLU (Acc ↑) | GSM8K (Acc ↑) | HumanEval (Pass@1 ↑) |
| LLaDA | Non-watermarked | 0.648 | 0.797 | 0.427 | 0.594 | 0.775 | 0.360 |
| | KGW | 0.558 | 0.662 | 0.092 | 0.520 | 0.464 | 0.055 |
| | PATTERN-MARK | 0.570 | 0.635 | 0.134 | 0.532 | 0.438 | 0.073 |
| | dgMARK | **0.647** | **0.787** | **0.280** | **0.588** | **0.735** | **0.226** |
| | dgMARK +3-beam | **0.647** | 0.771 | 0.268 | 0.580 | 0.678 | 0.152 |
| LLaDA 1.5 | Non-watermarked | 0.650 | 0.821 | 0.400 | 0.601 | 0.808 | 0.348 |
| | KGW | 0.567 | 0.726 | 0.104 | 0.536 | 0.582 | 0.092 |
| | PATTERN-MARK | 0.579 | 0.670 | 0.152 | 0.540 | 0.513 | 0.079 |
| | dgMARK | **0.649** | **0.814** | **0.317** | **0.596** | **0.759** | **0.201** |
| | dgMARK +3-beam | **0.649** | 0.774 | 0.207 | 0.588 | 0.723 | 0.134 |
| Dream | Non-watermarked | 0.700 | 0.800 | 0.427 | 0.630 | 0.789 | 0.420 |
| | KGW | 0.558 | 0.661 | 0.287 | 0.523 | 0.444 | 0.134 |
| | PATTERN-MARK | 0.594 | 0.652 | 0.335 | 0.551 | 0.639 | 0.287 |
| | dgMARK | **0.695** | **0.746** | **0.470** | **0.647** | **0.686** | **0.390** |
| | dgMARK +3-beam | **0.695** | 0.701 | 0.342 | 0.636 | 0.648 | 0.262 |

are first sampled from $p_\theta(\cdot|y_\mathcal{I}, x)$, and decoding then selects among these sampled proposals according to the standard confidence-based rule. Our multinomial version follows the same mechanism, except that it gives preference to parity-matching proposals. Therefore, in both greedy and multinomial settings, dgMARK can *implicitly* affect the final sequence distribution through proposal selection and decoding order, but it does not *directly* alter the learned conditional predictor itself. In this sense, dgMARK differs from watermarking methods that explicitly perturb logits or token probabilities: the watermark signal is introduced by a parity-aware preference rule layered on top of the standard dLLM decoding procedure.

**Text Generation Quality.** Table 3 reports benchmark results on MMLU, GSM8K, and HumanEval under greedy and multinomial sampling, including (1) the non-watermarked baseline, (2) KGW, (3) PATTERN-MARK, (4) dgMARK, and (5) dgMARK with 3-beam search. For KGW and PATTERN-MARK, we set $\gamma = 0.5$ and $\delta = 3$, chosen to match the detectability of dgMARK with 3-beam search based on Table 2. Across benchmarks, watermarking induces only minor degradation on MMLU and GSM8K, while HumanEval exhibits a larger drop, consistent with the low entropy of code generation tasks (Lee et al., 2024). Among all methods, dgMARK yields the smallest quality loss. As shown in Table 2, the perplexity increase of dgMARK over the non-watermarked baseline is consistently smaller than that of KGW and PATTERN-MARK. This suggests that *controlling the decoding order* provides a less

intrusive watermarking mechanism than *manually altering token probabilities*, achieving comparable detectability with better quality preservation.

**Robustness to Post-editing.** We evaluate robustness under both token-level edits and paraphrasing attacks. For token-level attacks, we apply random insertions, deletions, and substitutions with budgets $\epsilon \in \{0.1, 0.2, 0.3, 0.4\}$ and use sliding-window detection with window size $w = 8$. Figure 2 reports ROC curves and shows that dgMARK remains effective even under heavy perturbations (e.g., $\epsilon = 0.4$). Additional comparisons with KGW and PATTERN-MARK are provided in the Appendix.

For paraphrasing attacks, we consider two settings. First, we use DIPPER-11B (Krishna et al., 2023), a paraphrase-generation model commonly used to stress-test watermark detectors: DIPPER-1 paraphrases with a fixed lexical-modification ratio, and DIPPER-2 further adds 10% order diversity. Second, we evaluate paraphrasing by Llama 3-8B (Instruct) (Grattafiori et al., 2024) using a prompt adapted from Kirchenbauer et al. (2024) (the exact prompt is provided in the Appendix). As shown in Figure 3, detectability decreases as paraphrasing intensity increases for all methods, but dgMARK, especially with 3-beam search, remains reliably detectable despite producing lower-PPL text.

## 5. Practical Considerations

This section discusses practical factors that affect the deployment of dgMARK in real systems, beyond the core results

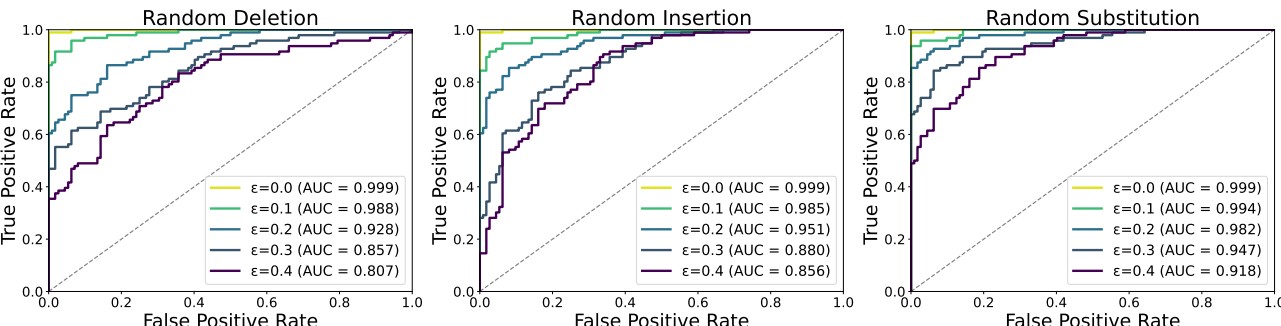

*Figure 2.* **ROC curves under post-editing attacks.** Illustration of the sliding-window strategy against random deletion, insertion, and substitution with modification budget $\epsilon$. Watermarks are generated by standard dgMARK ($k = 1$) using multinomial sampling.

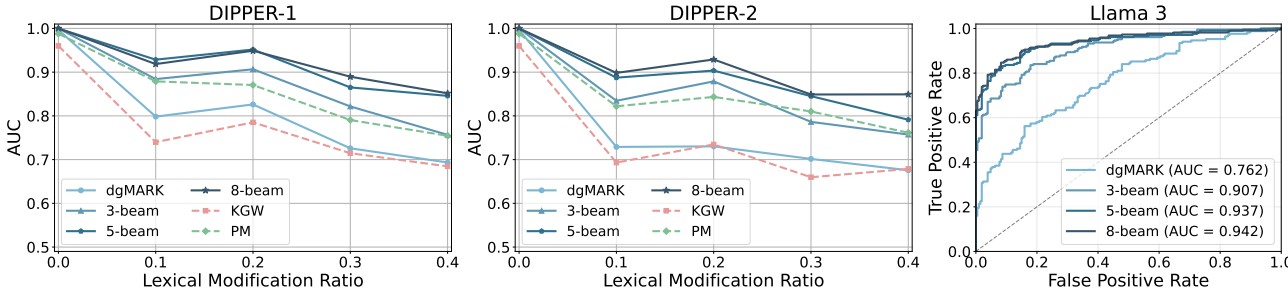

*Figure 3.* **Detection AUC under paraphrasing attacks.** Results for dgMARK with DIPPER (Krishna et al., 2023): (Left) paraphrasing at predefined ratios via lexical modification; (Middle) paraphrasing with ratio-adjusted lexical modification and an additional 10% order diversity. Comparative results with KGW and PATTERN-MARK are included to assess relative robustness. (Right) paraphrasing generated by Llama 3-8B (Instruct), evaluated using ROC curves.

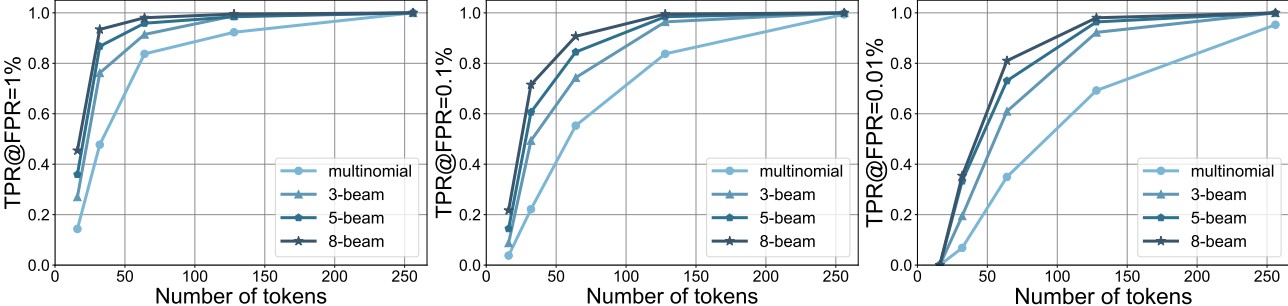

*Figure 4.* **Watermark detectability vs. sequence length.** Results under multinomial sampling with beam sizes $k \in \{1, 3, 5, 8\}$, reported as TPR at FPR levels of 10%, 1%, 0.1%, and 0.01%. Generation lengths are set to $\{16, 32, 64, 128, 256\}$, where the 256 setting includes sequences with length $\geq 200$.

in Section 4. In particular, we examine how detectability scales with output length and how block-wise generation (i.e., a common strategy for efficient long-form decoding in dLLMs) interacts with watermark embedding.

**Generation Length.** Since dgMARK is detected through a statistical test on parity matches, longer outputs provide more evidence and typically improve reliability. We quantify this effect by reporting TPR at several FPR levels (10%, 1%, 0.1%, 0.01%) for $n \in \{16, 32, 64, 128, 256\}$ under multinomial sampling. As shown in Figure 4, detectability improves steadily with length. In particular, when $n \geq 200$,

dgMARK with beam size $k \geq 3$ consistently achieves TPR = 1.0 even at FPR = 0.01. This suggests that watermark detection is most reliable for long-form generations (e.g., summaries, reports, and stories), while very short outputs may require either aggregation across multiple responses or a less stringent operating point.

**Block-wise Generation.** For efficiency and stability over long contexts, dLLMs are often decoded in blocks (Arriola et al., 2025; Nie et al., 2025). To assess the impact of this design choice, we generate texts with a fixed target length of 256 tokens using block sizes $\{8, 16, 32, 64, 128\}$ and

*Table 4.* **Effect of block length on watermarking.** TPR at FPR levels of 10%, 1%, 0.1%, and 0.01%, along with PPL on the C4 dataset for dgMARK with block lengths of 8, 16, and 32.

| Sampling | Block | PPL | TPR@FPR | | | |
|---|---|---|---|---|---|---|
| | | | 10% | 1% | 0.1% | 0.01% |
| Multinomial | 8 | 4.98 | 100.00 | 97.69 | 93.06 | 80.56 |
| | 16 | 5.16 | 99.50 | 98.02 | 97.03 | 89.60 |
| | 32 | 5.27 | 100.00 | 100.00 | 99.41 | 95.29 |
| Greedy | 8 | 4.28 | 93.42 | 74.12 | 48.25 | 29.82 |
| | 16 | 4.42 | 96.77 | 86.64 | 69.59 | 50.23 |
| | 32 | 4.44 | 97.86 | 91.98 | 76.47 | 60.96 |

report TPR@FPR and perplexity in Table 4. Because block sizes of 64 and 128 frequently fail to produce sequences longer than 200 tokens, we report TPR@FPR only up to block size 32; the appendix provides the resulting effective sequence lengths under each setting. Overall, larger block sizes tend to strengthen watermark embedding (i.e., higher detectability at the same threshold), while excessively large block sizes reduce generation stability for long sequences. In practice, moderate block sizes (e.g., 16 or 32) provide a favorable balance between efficiency, stable long-form generation, and watermark reliability.

**Additional Analyses.** Additional results in Appendix E include comparisons with a concurrent dLLM watermarking baseline, GPT-as-judge imperceptibility evaluation, mod-$m$ watermark-strength ablations, Fast-dLLM-style multi-token decoding, and expanded security/robustness discussions.

## 6. Conclusion

We introduced dgMARK, a decoding-guided watermarking method for discrete diffusion language models (dLLMs). Instead of biasing token probabilities, dgMARK embeds watermark signals by guiding the decoding order, without directly reweighting the model's learned conditional probabilities. Comprehensive experiments demonstrate that dgMARK provides strong detectability, minimal quality degradation, and robustness against post-editing. These results establish decoding-based watermarking as an effective and practical approach for ensuring provenance in dLLMs.

## Acknowledgements

This work was supported in part by Institute of Information & communications Technology Planning & Evaluation (IITP) grant funded by the Korea government (MSIT) (No. RS-2024-00457882, AI Research Hub Project), IITP grant funded by the Korean Government (MSIT) (No. RS-2020-II201361, Artificial Intelligence Graduate School Program (Yonsei University)), and the National Research Foundation of Korea (NRF) grant funded by the Korea government (MSIT) (No. RS2025-23525649).

## Impact Statement

This paper investigates watermarking techniques for text generated by discrete diffusion language models, with the goal of further advancing the field of machine learning. There are many potential societal consequences of our work, none of which we feel must be specifically highlighted here.

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

## A. LLM Usage

This manuscript made limited use of Large Language Models (LLMs) for language editing only. Their role was restricted to improving readability—such as grammar, style, and flow—without contributing to the conception of ideas, analyses, or results. All scientific content remains the original work of the authors, who carefully reviewed any edited text to ensure accuracy and integrity.

## B. Reproducibility

For reproducibility, Table 5 lists the external resources employed in our experiments, along with their corresponding licenses and references.

*Table 5.* **List of external resources.** Resources used in the experiments, with corresponding licenses and references.

| Resource | License | Reference |
| --- | --- | --- |
| LLaDA Instruct | MIT License | Nie et al. (2025) |
| LLaDA 1.5 | MIT License | Zhu et al. (2025) |
| LLaDA 2.0-mini | Apache License 2.0 | Bie et al. (2025) |
| Dream Instruct | Apache License 2.0 | Ye et al. (2025) |
| Gemma3 | Gemma | Team et al. (2025) |
| Meta-Llama-3-8B | Llama3 | Grattafiori et al. (2024) |
| Llama-3.3-70B-Instruct-AWQ[1] | Llama3.3 | Grattafiori et al. (2024) |
| Dipper Paraphraser | Apache License 2.0 | Krishna et al. (2023) |
| C4 | ODC-BY | Raffel et al. (2023) |
| WritingPrompts | MIT License | Fan et al. (2018) |
| MMLU | MIT License | Hendrycks et al. (2021) |
| GSM8K | MIT License | Cobbe et al. (2021) |
| HumanEval | MIT License | Chen et al. (2021) |

## C. dgMARK with Beam Search Algorithm

Algorithm 3 summarizes the complete procedure of Top-$k$ one-step lookahead beam search.

---

**Algorithm 3** dgMARK with Top-$k$ One-Step Lookahead

---

**Require:** Prompt $x$; output length $n$; predictor $p_\theta$; decoding strategy $\mathcal{F}$; watermark key $\xi$; beam size $k$

1: $y \leftarrow [\text{MASK}]^n$;    $\mathcal{I} \leftarrow \emptyset$
2: **function** NEXTMATCHCOUNT$(y, \mathcal{I})$
3:     Compute $(\hat{r}_\ell, \hat{v}_\ell)$ for all $\ell \notin \mathcal{I}$ using $\mathcal{F}(\ell; p_\theta, x, y_{\mathcal{I}})$
4:     **return** $\sum_{\ell \notin \mathcal{I}} \mathbb{1}[\hat{v}_\ell \in \mathcal{G}_\ell]$
5: **end function**
6: **for** $i = 1, \ldots, n$ **do**
7:     Compute $(r_j, v_j)$ for all $j \notin \mathcal{I}$ using $\mathcal{F}(j; p_\theta, x, y_{\mathcal{I}})$
8:     $\mathcal{C} \leftarrow \{j \notin \mathcal{I} | v_j \in \mathcal{G}_j\}$
9:     **if** $\mathcal{C} = \emptyset$ **then** $\mathcal{C} \leftarrow \{j \notin \mathcal{I}\}$
10:    **end if**
11:    $\mathcal{T} \leftarrow$ indices of the top-$k$ elements of $\mathcal{C}$ by $r_j$
12:    $k^\star \leftarrow \arg\max_{j \in \mathcal{T}}$ NEXTMATCHCOUNT$(y$ with $y_j \leftarrow v_j, \mathcal{I} \cup \{j\})$
13:    $y_{k^\star} \leftarrow v_{k^\star}$;    $\mathcal{I} \leftarrow \mathcal{I} \cup \{k^\star\}$
14: **end for**
15: **Return** $y$

---

[1]https://huggingface.co/kosbu/Llama-3.3-70B-Instruct-AWQ

# D. Experiment Details

## D.1. Baselines

We evaluate watermarking performance using PATTERN-MARK and KGW, which represent watermarking schemes for order-agnostic and autoregressive-style generation, respectively. PATTERN-MARK operates independently of token generation order, whereas KGW requires a left-to-right decoding process. Accordingly, when applying KGW to dLLMs, we generate tokens sequentially in a left-to-right manner and use 1-gram tokens as watermark keys for embedding. For KGW, we use $\gamma = 0.5$. For PATTERN-MARK, we set $\gamma = 0.5$ and use two alternating patterns, $(0, 1)$ and $(1, 0)$, which induce an alternating color (green/red) assignment across token positions. In our experiments, the deterministic function $f$ maps each token to a binary value based on the token ID modulo 2. Although we use a simple modulo operation for the mapping in our experiments, the framework is designed to be compatible with any cryptographically secure pseudorandom functions (PRFs). By employing a PRF for $f(v, \xi)$, the resulting binary assignment becomes computationally infeasible to predict without knowledge of the secret key $\xi$, thereby enhancing the resilience of the watermark against adversarial reverse-engineering.

## D.2. Evaluation Details

For benchmark evaluation, we adopted a block-wise text generation strategy. We utilized the authors' original implementations for the LLaDA family and followed prior studies to implement the same strategy for Dream-7B. Table 6 summarizes the block and total sequence lengths used in our evaluation. The confidence strategy was applied across all benchmarks. The non-watermarked baselines for the LLaDA family were evaluated using the block lengths specified in their original papers, while Dream-7B was evaluated using the block lengths reported in Table 6.

## D.3. Hardware Specification

The experiments were conducted under the following hardware configurations: (1) Text generation: Non-watermarked and watermarked text generated on NVIDIA GeForce RTX 4090, while text generation for LLaDA 2.0 was performed on an NVIDIA L40S. (2) Text perplexity (PPL) computation: Performed on NVIDIA GeForce RTX 5090. (3) Benchmark evaluations: The LLaDA family was evaluated on RTX 5090, Dream-7B evaluated on an RTX 4090.

*Table 6.* **Inference configurations.** A block length shorter than the total length indicates the use of the block-wise generation strategy for LLaDA-8B, LLaDA 1.5-8B, and Dream-7B.

|  | LLaDA-8B | | LLaDA 1.5-8B | | Dream-7B | |
|---|---|---|---|---|---|---|
|  | Block Length | Total Length | Block Length | Total Length | Block Length | Total Length |
| MMLU | 3 | 3 | 3 | 3 | 3 | 3 |
| GSM8K | 8 | 256 | 16 | 256 | 32 | 256 |
| HumanEval | 8 | 512 | 8 | 512 | 32 | 512 |

## D.4. Llama Paraphrase Attack

For paraphrasing in robustness evaluations of watermarking, we use LLaMA 3-8B (Instruct) with a task-specific instruction to generate paraphrases. The model is queried with the following prompt, using a sampling temperature of 0.2 and a maximum token limit of 256. The prompt is adapted from prior work (Kirchenbauer et al., 2024).

---
**Llama 3 Prompt**

As an expert copy-editor, please rewrite the following text in your own voice while ensuring that the final output contains the same information as the original text and has roughly the same length. Please paraphrase all sentences and do not omit any crucial details. Additionally, please take care to provide any relevant information about public figures, organizations, or other entities mentioned in the text to avoid any potential misunderstandings or biases. Respond only with the paraphrased text.

---

# E. Additional Results

## E.1. Computational Overhead

We measured the per-token decoding time of dgMARK, and Table 7 shows that multinomial sampling with $k = 1$ introduces negligible overhead relative to standard decoding. Increasing the beam size to $k = 3$ yields a substantial improvement in detectability while increasing cost by only $\approx 2.7\times$. Larger beam sizes naturally incur additional overhead, as beam search evaluates multiple candidate sequences in parallel.

*Table 7.* **Computational overhead.** Comparison of ms/token and overhead between the non-watermark baseline and dgMARK with beam sizes $k \in \{1, 3, 5\}$

| Method | Non-watermarked | Watermarked | | |
|---|---|---|---|---|
| | | dgMARK | +3 beam | +5 beam |
| ms / token | 60.52 | 69.95 | 165.50 | 229.69 |
| Overhead | $1.00\times$ | $1.16\times$ | $2.73\times$ | $3.80\times$ |

## E.2. Watermark Detectability

**Results on LLaDA 2.0.** Table 8 compares the watermark detectability of dgMARK with baselines, KGW and PATTERN-MARK, under different watermark strengths ($\delta \in \{1, 2, 3\}$). The results indicate that dgMARK with 3-beam search consistently achieves higher detectability while maintaining lower PPL compared to existing watermarking methods.

**Evaluation with Additional Decoding Strategies.** Tables 9 and 10 summarize the experimental results of dgMARK with entropy and margin-based decoding strategies, respectively. Under the entropy strategy, increasing the beam size $k$ led to substantially stronger watermark detectability with only a marginal increase in PPL. In contrast, under the margin strategy, the PPL increase was negligible at $k = 1$ but grew considerably for $k \geq 3$, while detectability remained high. Moreover, Table 9 demonstrates that watermark embedding is more effective under multinomial sampling than greedy sampling.

**Evaluation on an Additional Dataset.** Table 11 reports results on the Writing Prompts dataset with LLaDA 1.5-8B. Across prompts inducing diverse writing styles, dgMARK consistently achieved higher watermark detectability as the beam size $k$ increased. Notably, dgMARK incurred only a negligible PPL penalty, indicating effective watermarking with minimal impact on text quality.

**Additional Comparison with dLLM Watermarking Baselines.** We additionally compare dgMARK against a concurrent dLLM watermarking baseline (Gloaguen et al., 2026) across both watermark detectability and downstream benchmark performance. The baseline follows the sampling configuration reported in the original work, which employs multinomial sampling with a default temperature of 0.5. Empirically, fully deterministic greedy sampling in this setting resulted in unstable text generation. We therefore report dgMARK results under both greedy and multinomial sampling settings. In addition, we include a beam search variant of dgMARK with beam size 3 to further evaluate the trade-off between watermark detectability and downstream task performance. As shown in Table 12, dgMARK achieves competitive performance compared to the concurrent dLLM watermarking baseline across both detectability and downstream benchmark evaluations. In particular, stronger watermark settings improve detectability, while lower-entropy reasoning tasks such as GSM8K and HumanEval exhibit larger performance degradation under stronger watermark enforcement. These observations are consistent with prior findings that code and mathematical reasoning benchmarks remain challenging settings for watermarking methods due to their strict syntactic and logical constraints.

## E.3. Text Generation Quality

**Additional Perplexity Analysis.** Figure 8 illustrates the PPL comparison between non-watermarked text and text watermarked using dgMARK under the entropy strategy. The results demonstrate that dgMARK is compatible with various decoding strategies while incurring minimal quality degradation.

**Additional Qualitative Results.** Illustrative examples in Figure 16 highlight the difference in parity-matching ratios between watermarked and non-watermarked text. Tables 16 to 18 present qualitative examples from LLaDA, LLaDA 1.5, and Dream under multinomial sampling, and highlight differences in the parity-matching ratio between non-watermarked and watermarked text.

**GPT-as-a-Judge Evaluation.** As a proxy for imperceptibility evaluation, we use the GPT-4o-mini API to compare the quality of watermarked and non-watermarked text. The evaluation uses an instruction-based prompt adapted from prior works (Jovanović et al., 2024; Jiang et al., 2026b), directing the judge to disregard truncation artifacts and focus solely on linguistic fluency. As shown in Table 13, dgMARK maintains generation quality comparable to the non-watermarked baseline across all evaluation criteria, including coherence, clarity, and naturalness, under different beam size settings. The detailed evaluation prompt is specified in the following prompt configuration.

---

**GPT-as-a-judge Prompt**

You are an impartial expert evaluator of linguistic text quality.
The given text is a continuation generated from a truncated C4 sample. It may start or end abruptly because the generation length is fixed. Do NOT penalize truncation or incompleteness.
Evaluate ONLY linguistic quality on these criteria:
    - Coherence: logical flow of ideas
    - Clarity: easy to understand
    - Naturalness: how fluent / human-written the text appears
Rate each from 1 (poor) to 5 (excellent). Compute "overall" as the average of the three.
Return only a JSON object in exactly the following structure:
{
    "coherence": number,
    "clarity": number,
    "naturalness": number,
    "overall": number
}

---

### E.4. Empirical Trade-offs and Practical Extensions

**Mod-$m$ Extension.** The parity-based $(\bmod\ 2)$ construction used in the main formulation defines an alternating two-group matching scheme and can be naturally generalized to a modulo base $m$. Under this formulation, the vocabulary is partitioned into $m$ disjoint hash-based groups, and the target matching group varies according to the decoding position $j \pmod{m}$. The choice of $m$ therefore acts as a watermark-strength knob. Larger $m$ imposes a more selective matching condition, which can strengthen the watermark constraint, but the signal is also distributed across more groups; this may weaken the statistical bias per group and potentially reduce detectability when $m$ becomes too large. In the range tested here, mod 3 and mod 4 improve detectability relative to the default mod 2 setting, with only a moderate increase in perplexity, suggesting a practical quality–detectability trade-off (Table 14).

**Multi-token Generation.** To evaluate dgMARK under parallel decoding, we implement a Fast-dLLM-style multi-token decoding scheme (Wu et al., 2026) on top of LLaDA 1.5. The standard Fast-dLLM strategy performs parallel decoding by simultaneously committing all tokens whose confidence scores exceed 0.9, while decoding otherwise falls back to the single highest-confidence token. To integrate dgMARK, we adapt this scheme to operate over the target matching group, selecting all tokens within this set whose confidence scores exceed 0.9. If none satisfy this condition, decoding falls back to the single highest-confidence token within the matching group. As shown in Table 15, dgMARK successfully adapts to this parallelized setting, maintaining comparable text quality while retaining strong detectability. Despite a slight dilution of the watermark signal attributable to simultaneous token commitments, dgMARK remains highly effective in this practical, fast-decoding setup.

### E.5. Robustness against Post-editing Attacks

**Distributions of Parity-Matching Ratios.** Figures 5 to 7 show the distributions of window-level parity-matching ratios under random token insertion, deletion, and substitution attacks, respectively. The parity-matching ratio measures the proportion of tokens within a sliding window that satisfy the parity condition (i.e., $y_i \in \mathcal{G}_i$). For insertions and deletions, parity shifts cause multimodality in the matching ratios, providing an indicator of watermark presence. In contrast, under substitutions, the distribution of matching ratios tends to resemble the intact distribution of watermarked text. Consequently, across all three perturbation types, the distribution of matching ratios for watermarked text remains distinguishable from that of non-watermarked text.

**Comparison with Existing Methods.** Figure 9 presents ROC curves comparing dgMARK with existing watermarking schemes, including KGW and PATTERN-MARK. dgMARK outputs lower-PPL text while achieving detection performance comparable to existing methods and demonstrating notable robustness to random substitution attacks.

**Hyperparameter Sensitivity.** Figures 10 and 11 illustrate ROC curves for watermark detection against the DIPPER-1 and DIPPER-2 attack scenarios, using the sliding-window strategy with window sizes $w \in \{8, 16, 32\}$. The results indicate robustness regardless of window size, with a slight advantage for smaller windows. Figures 12 to 14 present ROC curves for watermark detection against random token insertion, deletion, and substitution attacks, evaluated at beam size $k \in \{3, 5, 8\}$. The results demonstrate that increasing $k$ consistently strengthens robustness and watermark detectability.

**Security Considerations.** Providing formal security guarantees against a well-resourced adversary remains an open challenge in LLM watermarking. In practice, however, several factors increase the difficulty of such attacks. Inferring the token-to-group mapping requires collecting a large volume of watermarked text, as the watermark signal is subtle and distributed throughout the generated output. Moreover, periodically rotating the hash function (e.g., every 16 tokens) can further hinder inference attacks by limiting the amount of consistent statistical evidence available to the adversary.

### E.6. Ablation on Generated Length

Figure 15 shows the distribution of sequence lengths generated from 300 prompts using the block-wise generation strategy. The target length was fixed at 256 tokens, with block sizes set to $\{8, 16, 32, 64, 128\}$. The generated sequence lengths were grouped into five bins (1–50, 51–100, 101–150, 151–200, and 201–256 tokens) to visualize the proportion of sequences in each range. The results suggest that block size influences the sequence lengths in both multinomial and greedy sampling. Smaller block sizes (e.g., 8 or 16) tend to yield a higher proportion of longer sequences, whereas larger block sizes (e.g., 64 or 128) often lead to most sequences clustering in the 1–50 token range, indicating frequent failures to generate long sequences.

## F. Limitation

dgMARK exploits the sensitivity of dLLMs to decoding order, a property that arises from imperfect training rather than a fundamental property of the masked diffusion framework. In theory, a perfectly trained dLLM would be invariant to the choice of decoding order, rendering order-based watermarking ineffective. Consequently, the detectability of dgMARK is not a theoretically guaranteed property, but rather an empirical one that depends on the imperfection of current models.

*Table 8.* **Watermark Detectability Comparison.** Empirical results under greedy and multinomial sampling with **LLaDA 2.0** on the C4 dataset, reporting perplexity (PPL) and detection metrics. Greedy and multinomial sampling represent the non-watermarked baselines.

| Method | PPL ↓ | FPR↓ | TNR↑ | TPR↑ | FNR↓ | TPR@FPR ↑ | | | |
|---|---|---|---|---|---|---|---|---|---|
| | | | | | | 10% | 1% | 0.1% | 0.01% |
| **Greedy Sampling** | 5.63 | - | - | - | - | - | - | - | - |
| KGW ($\delta = 1$) | 6.30 | 0.0 | 1.0 | 0.038 | 0.962 | 82.78 | 46.89 | 19.62 | 9.09 |
| KGW ($\delta = 2$) | 7.44 | 0.0 | 1.0 | 0.701 | 0.299 | 99.02 | 95.10 | 87.25 | 75.49 |
| KGW ($\delta = 3$) | 8.63 | 0.0 | 1.0 | 0.959 | 0.041 | 100.00 | 100.00 | 97.96 | 96.94 |
| PATTERN-MARK ($\delta = 1$) | 5.71 | 0.0 | 1.0 | 0.000 | 1.000 | 13.38 | 2.82 | 0.70 | 0.00 |
| PATTERN-MARK ($\delta = 2$) | 6.55 | 0.0 | 1.0 | 0.038 | 0.962 | 60.90 | 33.83 | 15.04 | 9.77 |
| PATTERN-MARK ($\delta = 3$) | 7.89 | 0.0 | 1.0 | 0.652 | 0.348 | 94.33 | 85.82 | 74.47 | 58.87 |
| dgMARK | **7.33** | 0.0 | 1.0 | 0.817 | 0.183 | 99.30 | 98.59 | 92.25 | 87.32 |
| + 3-beam | 8.15 | 0.0 | 1.0 | 0.980 | 0.020 | 100.00 | 100.00 | 100.00 | 98.00 |
| **Multinomial Sampling** | 6.48 | - | - | - | - | - | - | - | - |
| KGW ($\delta = 1$) | 8.03 | 0.0 | 1.0 | 0.034 | 0.966 | 83.17 | 51.44 | 26.44 | 6.37 |
| KGW ($\delta = 2$) | 9.06 | 0.0 | 1.0 | 0.787 | 0.213 | 99.53 | 97.16 | 92.89 | 82.94 |
| KGW ($\delta = 3$) | 10.56 | 0.0 | 1.0 | 0.975 | 0.025 | 100.00 | 100.00 | 98.99 | 98.49 |
| PATTERN-MARK ($\delta = 1$) | 9.40 | 0.0 | 1.0 | 0.000 | 1.000 | 19.42 | 4.32 | 0.00 | 0.00 |
| PATTERN-MARK ($\delta = 2$) | 10.70 | 0.0 | 1.0 | 0.031 | 0.969 | 76.40 | 51.55 | 27.33 | 16.15 |
| PATTERN-MARK ($\delta = 3$) | 13.76 | 0.0 | 1.0 | 0.409 | 0.591 | 98.25 | 91.23 | 81.87 | 68.42 |
| dgMARK | **9.26** | 0.0 | 1.0 | 0.979 | 0.021 | 100.00 | 100.00 | 99.31 | 98.61 |
| + 3-beam | 10.86 | 0.0 | 1.0 | 1.000 | 0.000 | 100.00 | 100.00 | 100.00 | 100.00 |

*Table 9.* **Error rates of watermarking.** Empirical results under greedy and multinomial sampling with LLaDA 1.5 with **entropy strategy** on the C4 dataset, reported across different $z$-score thresholds.

| Sampling | PPL ↓ | $z = 4.0$ | | | | $z = 5.0$ | | | |
|---|---|---|---|---|---|---|---|---|---|
| | | FPR | TNR | TPR | FNR | FPR | TNR | TPR | FNR |
| dgMARK (Greedy) | 4.51 | 0.0 | 1.0 | 0.511 | 0.489 | 0.0 | 1.0 | 0.216 | 0.784 |
| + 3-beam | 4.84 | 0.0 | 1.0 | 0.970 | 0.030 | 0.0 | 1.0 | 0.867 | 0.133 |
| + 5-beam | 5.02 | 0.0 | 1.0 | 0.987 | 0.013 | 0.0 | 1.0 | 0.970 | 0.030 |
| + 8-beam | 5.16 | 0.0 | 1.0 | 0.996 | 0.004 | 0.0 | 1.0 | 0.960 | 0.040 |
| dgMARK (Multinomial) | 6.39 | 0.0 | 1.0 | 0.995 | 0.005 | 0.0 | 1.0 | 0.985 | 0.015 |
| + 3-beam | 6.16 | 0.0 | 1.0 | 1.000 | 0.000 | 0.0 | 1.0 | 1.000 | 0.000 |
| + 5-beam | 6.42 | 0.0 | 1.0 | 1.000 | 0.000 | 0.0 | 1.0 | 1.000 | 0.000 |
| + 8-beam | 6.78 | 0.0 | 1.0 | 1.000 | 0.000 | 0.0 | 1.0 | 1.000 | 0.000 |

*Table 10.* **Error rates of watermarking.** Empirical results under greedy sampling with LLaDA 1.5 with **margin strategy** on the C4 dataset, reported across different $z$-score thresholds. As the margin strategy (Kim et al., 2025) assumes greedy token selection, we report results using greedy selection.

| Sampling | PPL ↓ | $z = 4.0$ | | | | $z = 5.0$ | | | |
|---|---|---|---|---|---|---|---|---|---|
| | | FPR | TNR | TPR | FNR | FPR | TNR | TPR | FNR |
| dgMARK (Greedy) | 4.40 | 0.0 | 1.0 | 0.601 | 0.399 | 0.0 | 1.0 | 0.282 | 0.718 |
| + 3-beam | 9.17 | 0.0 | 1.0 | 1.000 | 0.000 | 0.0 | 1.0 | 1.000 | 0.000 |
| + 5-beam | 14.77 | 0.0 | 1.0 | 1.000 | 0.000 | 0.0 | 1.0 | 1.000 | 0.000 |
| + 8-beam | 17.94 | 0.0 | 1.0 | 1.000 | 0.000 | 0.0 | 1.0 | 1.000 | 0.000 |

*Table 11.* **Error rates of watermarking.** Empirical results under greedy and multinomial sampling with LLaDA 1.5 on the **"Writing Prompts"**, reported across different $z$-score thresholds.

| Sampling | PPL ↓ | $z = 4.0$ | | | | $z = 5.0$ | | | |
|---|---|---|---|---|---|---|---|---|---|
| | | FPR | TNR | TPR | FNR | FPR | TNR | TPR | FNR |
| dgMARK (Greedy) | 5.37 | 0.0 | 1.0 | 0.223 | 0.777 | 0.0 | 1.0 | 0.058 | 0.942 |
| + 3-beam | 5.42 | 0.0 | 1.0 | 0.836 | 0.164 | 0.0 | 1.0 | 0.424 | 0.576 |
| + 5-beam | 5.72 | 0.0 | 1.0 | 0.951 | 0.049 | 0.0 | 1.0 | 0.684 | 0.316 |
| + 8-beam | 5.92 | 0.0 | 1.0 | 0.976 | 0.024 | 0.0 | 1.0 | 0.868 | 0.132 |
| dgMARK (Multinomial) | 6.34 | 0.004 | 0.996 | 0.733 | 0.267 | 0.0 | 1.0 | 0.235 | 0.765 |
| + 3-beam | 6.44 | 0.004 | 0.996 | 0.987 | 0.013 | 0.0 | 1.0 | 0.811 | 0.189 |
| + 5-beam | 6.48 | 0.0 | 1.0 | 0.995 | 0.005 | 0.0 | 1.0 | 0.966 | 0.034 |
| + 8-beam | 6.95 | 0.0 | 1.0 | 0.995 | 0.005 | 0.0 | 1.0 | 0.976 | 0.024 |

*Table 12.* **dLLM watermarking comparison.** Watermark detection performance on the C4 dataset and downstream benchmark performance on LLaDA 1.5 for different watermark strengths ($\delta \in \{1, 2\}$). Results for dgMARK are additionally reported under greedy sampling, multinomial sampling, and beam search decoding.

| Method | PPL ↓ | FPR↓ | TNR↑ | TPR↑ | FNR↓ | MMLU (Acc ↑) | GSM8K (Acc ↑) | HumanEval (Pass@1 ↑) |
|---|---|---|---|---|---|---|---|---|
| Gloaguen et al. ($\delta = 1$) | 4.87 | 0.000 | 1.000 | 0.484 | 0.516 | 0.586 | 0.733 | 0.232 |
| Gloaguen et al. ($\delta = 2$) | 6.04 | 0.000 | 1.000 | 0.917 | 0.083 | 0.535 | 0.644 | 0.128 |
| dgMARK (Greedy) | 4.44 | 0.000 | 1.000 | 0.540 | 0.460 | 0.649 | 0.814 | 0.317 |
| + 3-beam | 4.75 | 0.000 | 1.000 | 0.963 | 0.037 | 0.649 | 0.774 | 0.207 |
| dgMARK (Multinomial) | 5.27 | 0.000 | 1.000 | 0.929 | 0.071 | 0.596 | 0.759 | 0.201 |
| + 3-beam | 5.40 | 0.000 | 1.000 | 1.000 | 0.000 | 0.588 | 0.723 | 0.134 |

*Table 13.* **Imperceptibility evaluation.** GPT-as-judge evaluation comparing the linguistic quality of non-watermarked text and dgMARK with beam sizes $k \in \{1, 3, 5, 8\}$ under greedy sampling.

| Method | GPT-as-judge score | | | |
|---|---|---|---|---|
| | Coherence | Clarity | Naturalness | Overall |
| Non-watermarked | 4.63 | 4.65 | 4.62 | **4.63** |
| dgMARK | 4.65 | 4.64 | 4.61 | **4.63** |
| + 3beam | 4.61 | 4.62 | 4.61 | 4.61 |
| + 5beam | 4.58 | 4.58 | 4.57 | 4.57 |
| + 8beam | 4.59 | 4.60 | 4.56 | 4.58 |

*Table 14.* **Mod-$m$ extension.** Comparison of detectability and perplexity across dgMARK variants on the C4 dataset with LLaDA 1.5 under greedy sampling, including the default parity-based setting ($m = 2$), generalized modulo extensions ($m \in \{3, 4\}$), and beam search decoding with beam size 3, relative to the non-watermarked baseline.

| Method | PPL ↓ | FPR↓ | TNR↑ | TPR↑ | FNR↓ | TPR@FPR ↑ | | | |
|---|---|---|---|---|---|---|---|---|---|
| | | | | | | 10% | 1% | 0.1% | 0.01% |
| **Non-watermarked** | 4.03 | - | - | - | - | - | - | - | - |
| dgMARK | 4.44 | 0.000 | 1.000 | 0.540 | 0.460 | 97.86 | 91.98 | 76.47 | 60.96 |
| dgMARK + 3beam | 4.75 | 0.000 | 1.000 | 0.963 | 0.037 | 100.00 | 99.54 | 98.62 | 97.25 |
| dgMARK (mod 3) | 4.56 | 0.005 | 0.995 | 0.782 | 0.218 | 98.32 | 95.53 | 91.06 | 82.68 |
| dgMARK (mod 3) + 3beam | 4.98 | 0.005 | 0.995 | 0.966 | 0.034 | 100.00 | 100.00 | 98.52 | 98.03 |
| dgMARK (mod 4) | 4.84 | 0.000 | 1.000 | 0.945 | 0.055 | 99.31 | 98.62 | 98.62 | 94.48 |
| dgMARK (mod 4) + 3beam | 5.06 | 0.000 | 1.000 | 0.989 | 0.011 | 100.00 | 100.00 | 98.92 | 98.92 |

*Table 15.* **Multi-token generation per step.** Compatibility of dgMARK with Fast-dLLM-style parallel decoding scheme, reporting perplexity (PPL) and detection metrics under greedy sampling on the C4 dataset with LLaDA 1.5. Non-watermarked baselines are included for both standard and fast dLLM decoding settings.

| Method | PPL ↓ | FPR↓ | TNR↑ | TPR↑ | FNR↓ | TPR@FPR ↑ | | | |
|---|---|---|---|---|---|---|---|---|---|
| | | | | | | 10% | 1% | 0.1% | 0.01% |
| Non-watermarked | 4.03 | - | - | - | - | - | - | - | - |
| dgMARK | 4.44 | 0.000 | 1.000 | 0.540 | 0.460 | 97.86 | 91.98 | 76.47 | 60.96 |
| Non-watermarked (fast dLLM) | 4.01 | - | - | - | - | - | - | - | - |
| dgMARK (fast dLLM) | 4.42 | 0.000 | 1.000 | 0.502 | 0.498 | 96.77 | 88.02 | 77.42 | 57.60 |

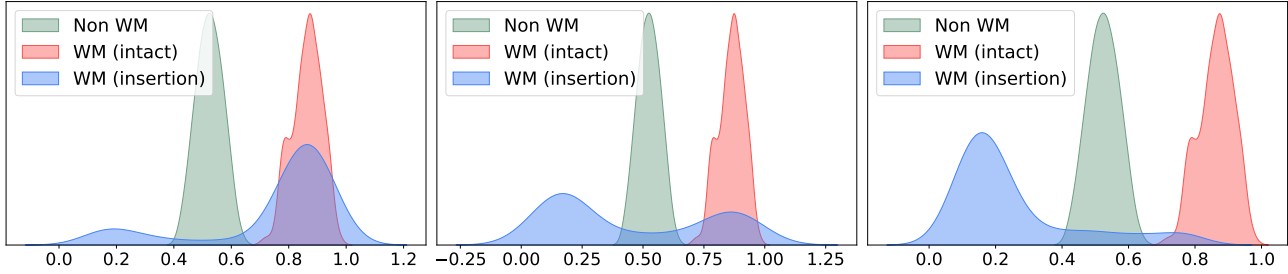

*Figure 5.* **Illustration of the distribution of parity alignment.** At window size $w = 32$, comparison of (1) non-watermarked texts (Non WM), (2) intact watermarked texts (WM), and (3) watermarked texts (WM) with **"random token insertions"**, where the number of inserted tokens increases from left to right. The x-axis represents the parity-matching ratio, defined as the proportion of tokens satisfying the parity condition, computed over a sliding window.

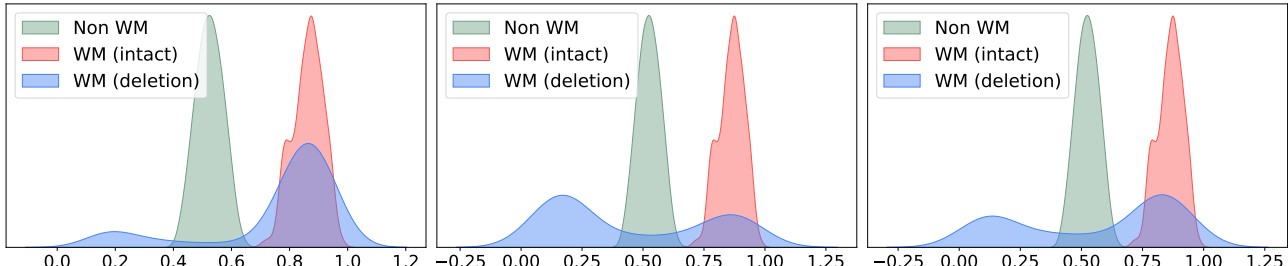

*Figure 6.* **Illustration of the distribution of parity alignment.** At window size $w = 32$, comparison of (1) non-watermarked texts (Non WM), (2) intact watermarked texts (WM), and (3) watermarked texts (WM) with **"random token deletion"**, where the number of deleted tokens increases from left to right. The x-axis represents the parity-matching ratio, defined as the proportion of tokens satisfying the parity condition, computed over a sliding window.

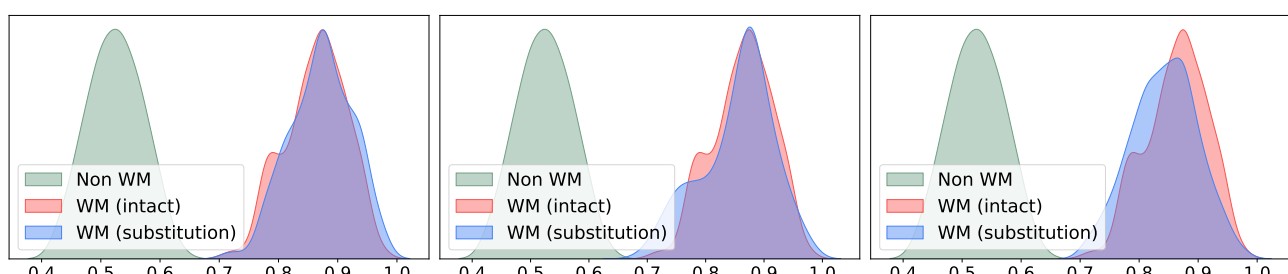

*Figure 7.* **Illustration of the distribution of parity alignment.** At window size $w = 32$, comparison of (1) non-watermarked texts (Non WM), (2) intact watermarked texts (WM), and (3) watermarked texts (WM) with **"random token substitution"**, where the number of substituted tokens increases from left to right. The x-axis represents the parity-matching ratio, defined as the proportion of tokens satisfying the parity condition, computed over a sliding window.

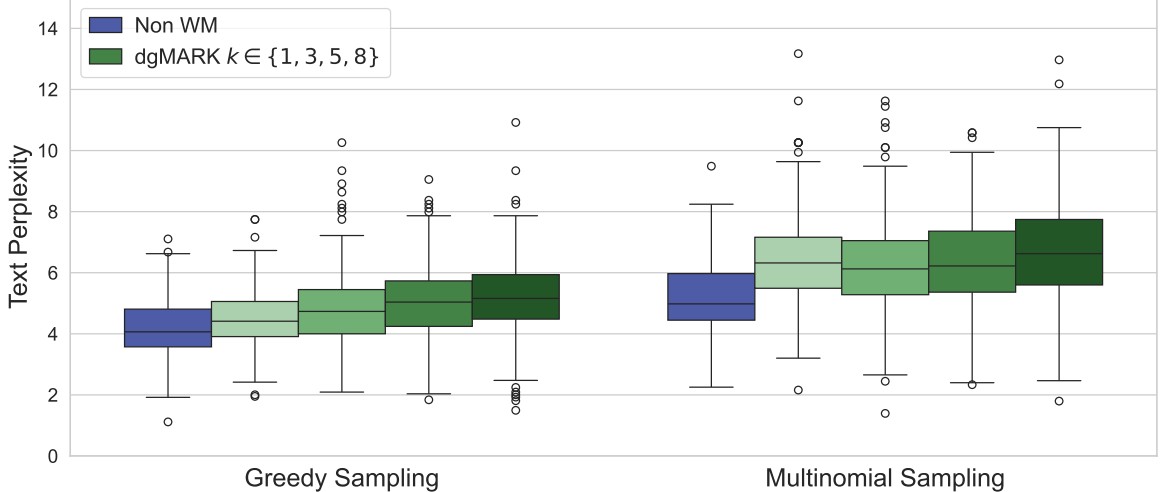

*Figure 8.* **Comparison of text perplexity using the "entropy strategy":** (1) Non-watermarked texts (Non WM) and (2) Watermarked texts generated by dgMARK with beam sizes $k \in \{1, 3, 5, 8\}$. Lighter green represents $k = 1$ and darker green represents $k = 8$.

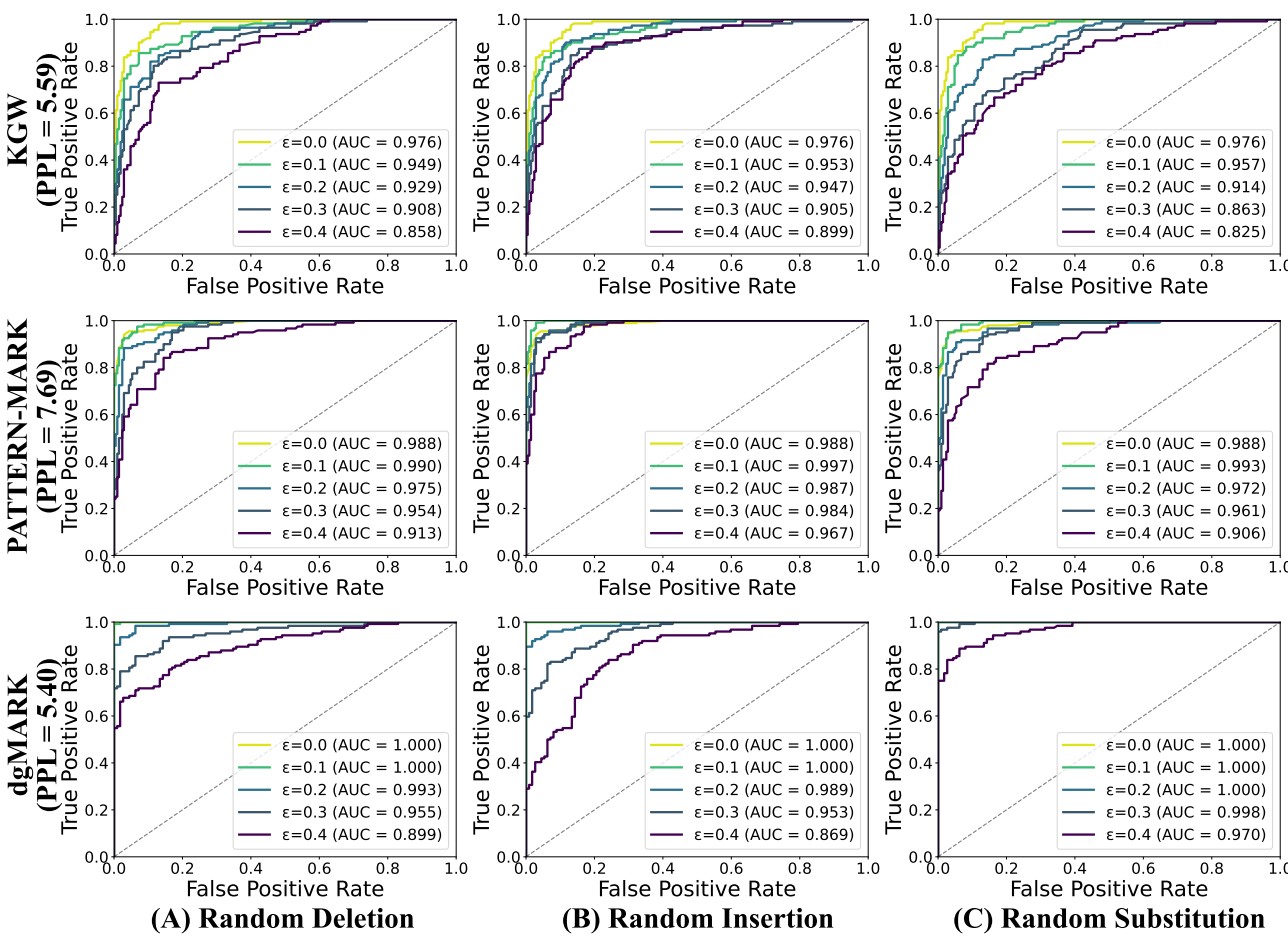

*Figure 9.* **ROC curves under post-editing attacks.** Illustration of the sliding-window strategy against (A) random deletion, (B) insertion, and (C) substitution with modification budget $\epsilon$. The comparison includes (1) KGW, (2) PATTERN-MARK, and (3) dgMARK with 3-beam search.

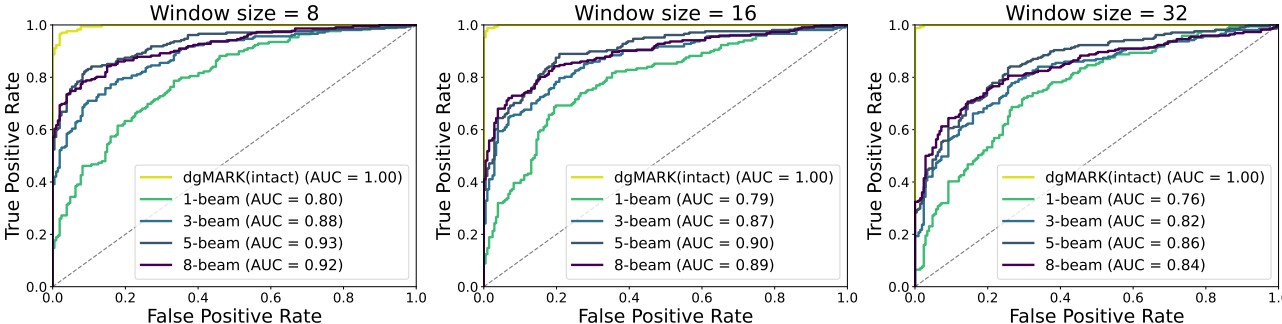

*Figure 10.* **ROC curves under the "DIPPER-1" setting.** Illustration of the sliding-window strategy for detection performance against paraphrasing attacks, evaluated at window sizes $w \in \{8, 16, 32\}$.

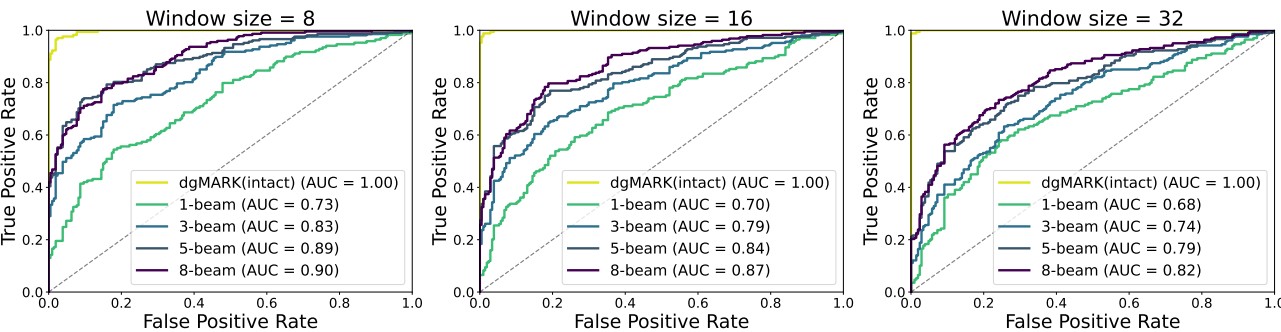

*Figure 11.* **ROC curves under the "DIPPER-2" setting.** Illustration of the sliding-window strategy for detection performance against paraphrasing attacks, evaluated at window sizes $w \in \{8, 16, 32\}$.

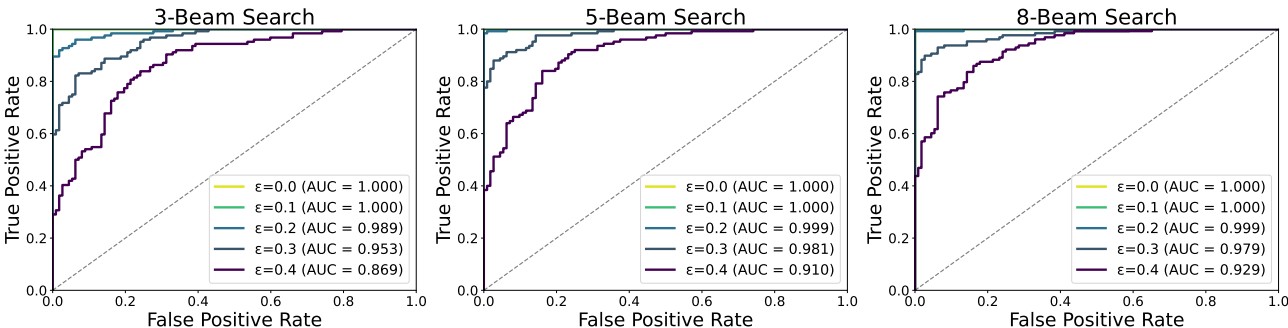

*Figure 12.* **ROC curves under post-editing attacks.** Illustration of the sliding-window strategy against **"random token insertion"** attacks with modification budget $\epsilon$, when texts are generated with beam sizes $\{3, 5, 8\}$.

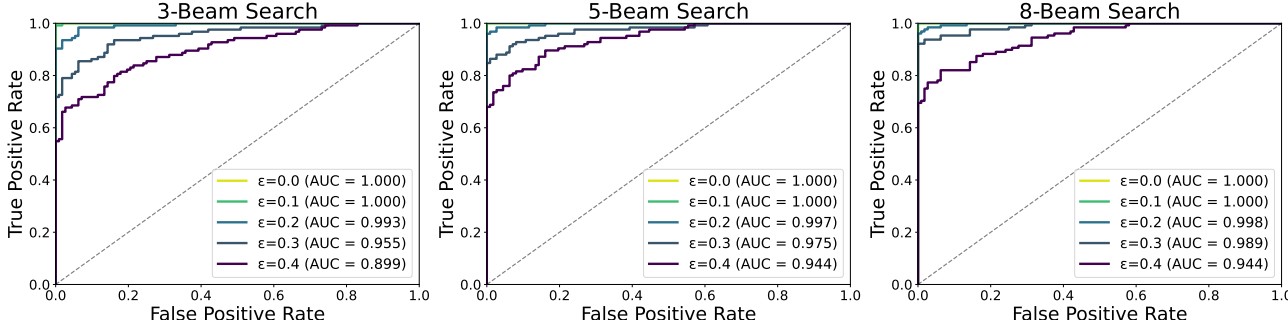

*Figure 13.* **ROC curves under post-editing attacks.** Illustration of the sliding-window strategy against **"random token deletion"** attacks with modification budget $\epsilon$, when texts are generated with beam sizes $\{3, 5, 8\}$.

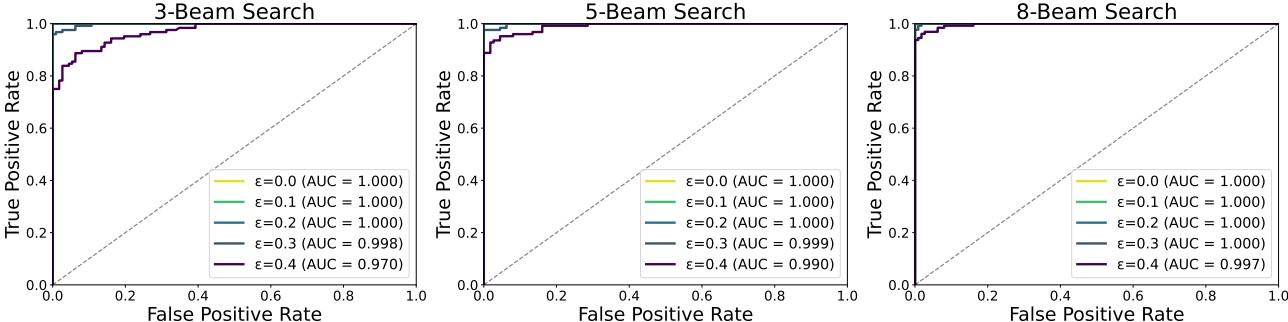

*Figure 14.* **ROC curves under post-editing attacks.** Illustration of the sliding-window strategy against **"random token substitution"** attacks with modification budget $\epsilon$, when texts are generated with beam sizes $\{3, 5, 8\}$.

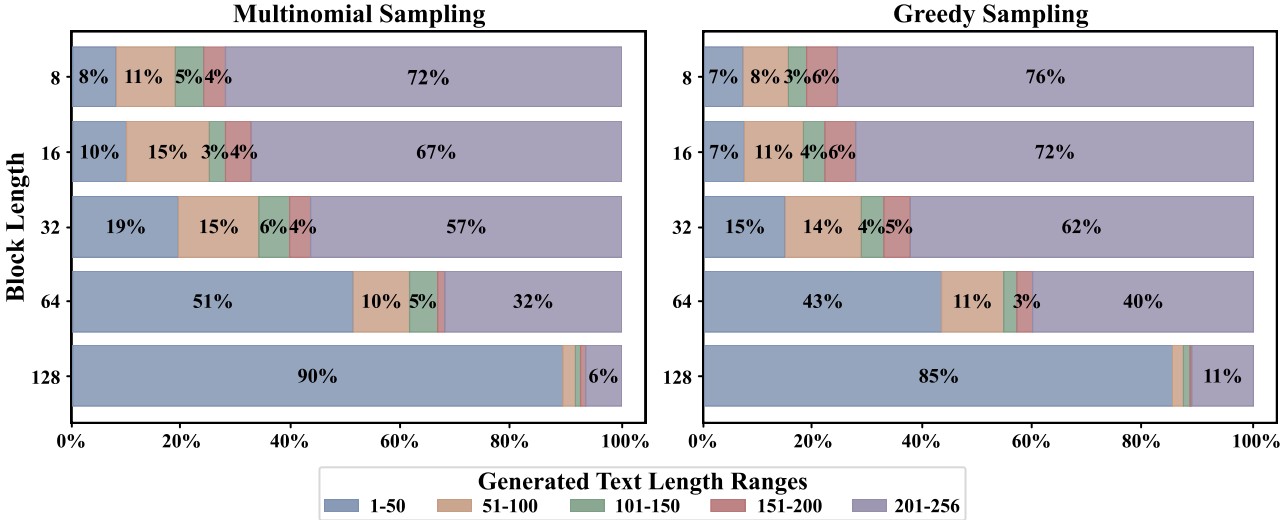

*Figure 15.* **Distribution of generated sequence lengths.** Text sequences are produced using the block-wise generation strategy with multinomial and greedy decoding under block sizes $\{8, 16, 32, 64, 128\}$.

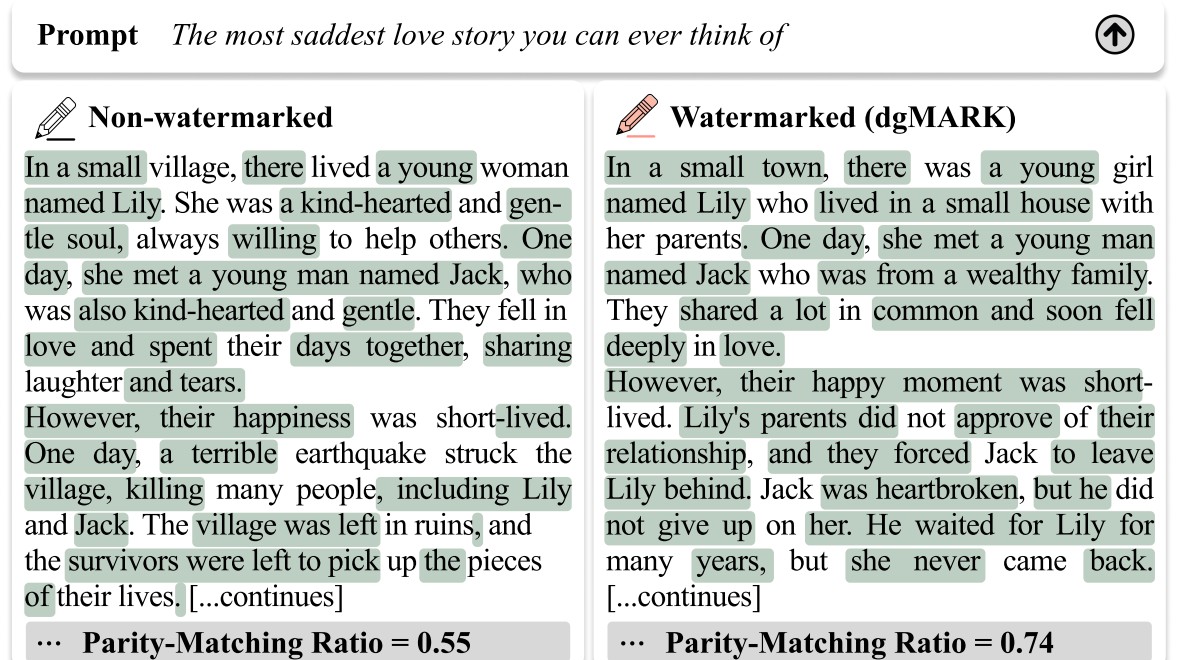

*Figure 16.* **Non-watermarked vs. watermarked text.** Generated by LLaDA 1.5 on the Writing Prompts dataset. Tokens highlighted in green indicate parity matches.

*Table 16.* **Non-watermarked vs. watermarked text.** Generated by **"LLaDA-8B"** on the Writing Prompts dataset with multinomial sampling, with parity-matching ratios reported for the outputs.

| Prompt | Non-watermarked | Watermarked |
|---|---|---|
| A kid doodling in a math class accidentally creates the world 's first functional magic circle in centuries. | That sounds like quite an adventure! The kid's doodling in a math class turned into a groundbreaking discovery. Can you imagine the excitement and wonder that must have filled the room? The teacher and classmates must have been in awe, trying to replicate the magic circle, and perhaps even asking the kid to explain how it works. [...continues] (***Matching Ratio = 0.54***) | That sounds like a fascinating and unexpected scenario! The idea of a child doodling in math class and accidentally stumbling upon a functional magic circle is intriguing. The concept of a "magic circle" is often associated with folklore and mythology, so the idea of such a discovery happening in a classroom setting is quite captivating. [...continues] (***Matching Ratio = 0.74***) |
| A person with a high school education gets sent back into the 1600s and tries to explain science and technology to the people. | That sounds like a fascinating scenario! A person with a high school education transported to the 1600s would likely face significant challenges in trying to explain science and technology to the people of the time. In the 1600s, science was still in its early stages, and most people had a limited understanding of the natural world. They relied heavily on religion, magic, and superstition to explain the world around them. The person would need to [...continues] (***Matching Ratio = 0.57***) | That sounds like an intriguing scenario! A person with a high school education transported to the 1600s would likely face many challenges in explaining science and technology to the people of that time. The 1600s were a time of great religious and philosophical debates, and many people were still grappling with the mysteries of the universe. However, the person with a high school education would likely be able to provide a basic overview of scientific concepts [...continues] (***Matching Ratio = 0.70***) |
| You were born on a colonizing spacecraft on your way to a distant star . You are the middle generation of history ; born to late to have seen Earth , and to early to live to see your destination. | What a fascinating scenario! You were born on a colonizing spacecraft, which means you're part of a pioneering human civilization that's expanding beyond our home planet. As the middle generation of history, you're caught between the era of Earth and the era of your new home, a distant star. Imagine the challenges and opportunities that come with living on a spacecraft. You'd have to adapt to a new environment, learn a new language, and develop a new way of [...continues] (***Matching Ratio = 0.43***) | That's a fascinating scenario! As the middle generation of history, you would have witnessed the dawn of space travel and the establishment of a new civilization on a distant star system. Your lifetime would be marked by the challenges of adapting to a new environment, developing cutting-edge technology, and exploring the vastness of space. As a colonizing spacecraft, you would have been exposed to microgravity environments, artificial gravity, and the complexities of space travel. [...continues] (***Matching Ratio = 0.73***) |
| Narrate a game of Civilization as though it were excerpted from a history textbook. | The game of Civilization is a strategy simulation game that allows players to experience the development of ancient civilizations. The game begins with the establishment of a small settlement, and players must gather resources, build cities, and wage wars to expand their territory. The game allows players to develop different aspects such as agriculture, architecture, [...continues] (***Matching Ratio = 0.46***) | In 2500, the city of Nineveh was established on the banks of the Euphrates River. Under the leadership of Hammurabi, its king, the city grew rapidly and became a major power in the region ofumeria. The code of Hammurabi, a set of laws, became a model for other cities in the region. In 1800 BC, the city of Susa was established on the [...continues] (***Matching Ratio = 0.68***) |

*Table 17.* **Non-watermarked vs. watermarked text.** Generated by **"LLaDA 1.5-8B"** on the Writing Prompts dataset with multinomial sampling, with parity-matching ratios reported for the outputs.

| Prompt | Non-watermarked | Watermarked |
|---|---|---|
| A kid doodling in a math class accidentally creates the world 's first functional magic circle in centuries. | That sounds like a fascinating scenario! The idea of a kid doodling in math class and accidentally creating a functional magic circle is quite intriguing. It's a testament to the power of curiosity and the importance of encouraging students to explore and experiment with their interests. The magic circle, also [...continues] (***Matching Ratio = 0.55***) | That sounds like a fascinating and unexpected event! The idea of creating a functional magic circle is intriguing, especially in the context of mathematics and geometry. Magic circles, also known as false circles, are shapes that appear to be circles, but are actually composed of straight lines and contain straight angles. [...continues] (***Matching Ratio = 0.73***) |
| A person with a high school education gets sent back into the 1600s and tries to explain science and technology to the people. | That sounds like a fascinating scenario. A person with a high school education trying to explain science and technology to the people in the 1600s would likely face significant challenges, as the understanding and acceptance of scientific concepts were extremely limited at that time. The person would need to be patient, persistent, and creative in their approach. [...continues] (***Matching Ratio = 0.53***) | That sounds like an intriguing scenario! A person with a high school education trying to communicate science and technology to people in the 1600s would be quite a challenge. They would need to explain complex concepts like physics, mathematics, astronomy, and even biology in a way that is relevant and understandable to the people of the time. [...continues] (***Matching Ratio = 0.68***) |
| You were born on a colonizing spacecraft on your way to a distant star . You are the middle generation of history ; born to late to have seen Earth , and to early to live to see your destination. | What a fascinating scenario. You're the middle generation of history, born on a colonizing spacecraft on your way to a distant star. This is a unique and extraordinary experience. As a child, you would have grown up in a microgravity environment, surrounded by advanced technology and a diverse group of people from different backgrounds. You would have had access to the latest education, healthcare, and entertainment, [...continues] (***Matching Ratio = 0.50***) | That's a fascinating scenario! To be born on a colonizing spacecraft on the way to a distant star, and being the middle generation of history, would mean that you were born after the spacecraft left Earth but before it arrived at its destination. This would offer a unique perspective on both Earth and the distant planet. In this situation, you would likely have grown up in a place far away from Earth, experiencing firsthand the challenges [...continues] (***Matching Ratio = 0.78***) |
| Narrate a game of Civilization as though it were excerpted from a history textbook. | In the year 2000, the game of Civilization was released by Sid Meier's Civilization series. The game was played by two players, each of whom would control one of the ten different civilizations: China, Egypt, Greece, Rome, Spain, England, France, Germany, Russia, and the United States. The game was divided into six phases: Exploration, Expansion, Science, Culture, Religion, and Warfare. The game was played on a map that consisted of land, water, and cities. Players had to build their cities, [...continues] (***Matching Ratio = 0.53***) | In the game of Civilization, you take on the role of one of the ancient civilizations: Egypt, Greece, Rome, China, Rome, Egypt, or Inca. Each civilization has its unique characteristics, strengths, and weaknesses. The game begins with the establishment of your city, which is the foundation of your empire. You must then build structures to expand your empire, such as roads, walls, buildings, temples, markets, libraries, theaters, schools, universities, and government buildings. [...continues] (***Matching Ratio = 0.69***) |

*Table 18.* **Non-watermarked vs. watermarked text.** Generated by **"Dream-7B"** on the Writing Prompts dataset with multinomial sampling, with parity-matching ratios reported for the outputs.

| Prompt | Non-watermarked | Watermarked |
|---|---|---|
| A kid doodling in a math class accidentally creates the world 's first functional magic circle in centuries. | During math class, a student named Alex was quietly doodling in their notebook, trying to escape the monotony of the lesson. As they drew a simple circle, they began to notice something strange. The circle seemed to have a life of its own, shifting and changing as if it were alive. Curious, Alex began to experiment with the circle, discovering that it had unique properties and could be manipulated in various ways. [...continues] (***Matching Ratio = 0.59***) | Once upon a time, in a small town, there lived a young high school student named Tom. He wasn't particularly fond of math, but he had a knack for art and drawing. One day, while bored during his geometry class, he began doodling in his notebook. He started drawing different shapes and patterns, trying to come up with some unique and interesting designs. As he was drawing, [...continues] (***Matching Ratio = 0.70***) |
| A person with a high school education gets sent back into the 1600s and tries to explain science and technology to the people. | The 1600s were a time of great exploration and discovery in the fields of science and technology. During this time, people were exploring the mysteries of the natural world and trying to understand the workings of the universe. However, the understanding of science and technology was quite limited compared to what we know today. As a person with a high school education, you might find it challenging to explain science and technology [...continues] (***Matching Ratio = 0.42***) | During the 1600s, the concept of science and technology would have been foreign to many people, who still believed in the teachings of the Church and superstition. However, a person with a high school education would try to explain the basics of science and technology in a simple and straightforward manner. They would start by discussing the scientific method and the importance of observation, experimentation, and evidence in determining facts and theories. [...continues] (***Matching Ratio = 0.74***) |
| You were born on a colonizing spacecraft on your way to a distant star . You are the middle generation of history ; born to late to have seen Earth , and to early to live to see your destination . | As a middle generation on the colonizing spacecraft, I have had the privilege of witnessing the progress of our mission and the challenges we faced along the way. Growing up on the spacecraft, I learned the history of our journey and the importance of our mission to establish a new home for humanity on a distant star. I remember the days when I would look out the window and marvel at the vastness of the universe, [...continues] (***Matching Ratio = 0.51***) | As a middle generation on a colonizing spacecraft on our way to a distant star, I experience a unique blend of longing and curiosity. Growing up within the confines of spacecraft, I never got to witness the beauty of Earth, the vastness of the night sky, or the familiarity of my ancestral home. However, I am fortunate to have the opportunity to explore the unknown and witness the far reaches of our solar system. [...continues] (***Matching Ratio = 0.82***) |
| Narrate a game of Civilization as though it were excerpted from a history textbook. | In the year 4000, the dawn of the ancient era was marked by the rise of the first civilizations around the world. The game began with the establishment of the first city-states along the banks of the Euphrates and Tigris rivers in Mesopotamia. As the game progressed, the civilizations made advancements in agriculture, architecture, and trade, laying the foundation for the development of complex societies. [...continues] (***Matching Ratio = 0.52***) | In the heart of the ancient world, rival civilizations faced off in the pursuit of prosperity and dominance. The game of civilization was played in the arena of time, with each turn representing a chapter in the grand tapestry of history. As the game advanced, so did the complexities of technology, diplomacy, and warfare. The early game saw the rise of cities, growth of culture, and the pursuit of science and technology. [...continues] (***Matching Ratio = 0.83***) |

