# OpenReview forum: "dgMARK: Decoding-Guided Watermarking for Diffusion Language Models"
_ICML.cc/2026/Conference — ICML 2026 regular_

### Official Review · Reviewer_N4Ns · 2026-02-22

**Soundness:** 3
**Presentation:** 3
**Significance:** 2
**Originality:** 2
**Overall Recommendation:** 4
**Confidence:** 4

**Summary:**

This paper proposes dgMark, a decoding-guided watermarking method designed for diffusion-based LLMs (dLLMs). The method binds watermark information to the parity of each generation position, selects high-reward candidate tokens during decoding, and adopts a sliding-window mechanism for watermark detection.

**Compliance With Llm Reviewing Policy:**

Affirmed.

**Final Justification:**

Given that the authors have addressed my concerns, I am willing to raise my score.

**Key Questions For Authors:**

1. **Vocabulary partition strategy**. Is the vocabulary partition at each position fixed throughout generation? If so, it may intuitively increase the risk of generating repetitive or stylistically constrained text. In some cases, this could reduce PPL while not necessarily improving actual text quality. Could the authors clarify this design choice and its impact?
2. **Extension to mod 3 (or higher moduli)**. The method binds watermark bits via a mod 2 (parity-based) design. Could the authors extend this to mod 3 or other modular schemes? How would such changes affect detectability, robustness, and text quality?
3. **Decoding inconsistency (Algorithm 2 vs. experiments)**. Algorithm 2 uses an argmax operation, yet the experiments report both Greedy and Multinomial sampling. How is this implemented in practice?
4. **Robustness to misalignment**. Since the watermark is tightly coupled to token positions (via parity), any positional shift would alter the parity structure. How does the method maintain robustness under insertion, deletion, paraphrasing, or token-level misalignment attacks?
5. **Trade-off between text quality and detectability**. Table 3 reports downstream task performance but does not provide corresponding watermark detection rates. This makes it unclear whether improved text quality comes at the cost of reduced detectability. Could the authors provide more detailed results illustrating the trade-off between task performance and watermark strength?
6. **Multi-token generation per step**. Diffusion LLMs may generate multiple tokens per step. Has the method been evaluated in such settings? How would this affect watermark binding and detection, particularly under decoding strategies such as Beam Search?
7. **Comparison with prior work**. Have the authors considered comparing their method with [1] ? Such a comparison would provide a more meaningful reference point and strengthen the empirical evaluation.

**Reference**

[1] Watermarking Diffusion Language Models, Gloaguen et al., ICLR 2026

**Limitations:**

yes

**Strengths And Weaknesses:**

**Strengths**
1. The paper explores watermarking for diffusion LLMs, which is a relatively underexplored direction. This work provides potentially useful insights for non-autoregressive watermarking design.
2. The paper is generally well organized, and the figures and tables are clear and easy to understand.

**Weaknesses**
1. Lack of theoretical guarantees. The proposed method appears largely heuristic, and there is no theoretical analysis regarding detectability, robustness, or unbiasedness guarantees. Furthermore, the modification to the original decoding procedure seems relatively limited.
2. Insufficient experimental validation. The experiments are not comprehensive. Some important baselines are missing, and there is limited ablation or in-depth analysis of the method itself.

---

> ### Author Rebuttal · Authors · 2026-03-30
>
> We thank the reviewers for the helpful feedback and we address the concerns below.
>
> ---
> >[W1] Lack of Theoretical Guarantees
>
> While our approach is empirical, we respectfully highlight two key points:
> - **Theoretical Bounds**: Formal guarantees for dLLM decoding order remain an open problem. While strict unbiasedness cannot be guaranteed due to model approximation, our method crucially avoids explicitly reweighting token probabilities.
> - **Practical Strength**: We view the "modification to existing decoding" as a major advantage. By avoiding heavy architectural changes or complex probability manipulations, our method serves as a lightweight, plug-and-play module that leverages any-order decoding and seamlessly integrates with existing dLLMs.
>
> >[W2, Q5, Q7] Experimental Validation, Comparisons, and Trade-offs
>
> We address the concerns as follows:
> - **Baseline Selection**: We specifically chose KGW and PATTERN-MARK to provide clear reference points for both standard autoregressive and order-agnostic watermarking. Additionally, the Appendix provides comprehensive robustness evaluations under post-editing attacks.
> - **Concurrent Comparisons**: While recent diffusion-specific methods [1] are concurrent works (discussed in Related Work), we agree that direct comparison is valuable. We provide additional comparisons with [1] in our responses to Reviewers ovQc and iFBj, consistent with the conclusions in the main paper.
> - **Quality vs. Detectability**: This trade-off is explicitly analyzed in Table 2, which jointly reports PPL and detection metrics (FPR/FNR). Table 3 simply evaluates downstream task performance using the exact same, fixed settings from Table 2 without additional tuning, rather than exploring the trade-off itself.
>
> In the revision, we will clarify our baseline rationale, incorporate the new comparisons, and make our experimental setup and trade-off analysis more explicit.
>
> >[Q1] Vocabulary partition and text quality
>
> The vocabulary partition is defined by a balanced hash and varies across positions. Like standard watermarking strategies, our method employs a soft preference for parity-matching tokens rather than strictly forcing them. Intuitively, the model selects a preferred token only when it does not hurt the surrounding context or stylistic flow, remaining free to choose non-matching tokens when appropriate. This introduces a detectability–quality trade-off commonly observed in watermarking schemes, but our soft preference empirically preserves expressiveness and text quality without noticeable repetitiveness.
>
> >[Q2, Q6] Extensions to modular schemes and parallel decoding
>
> - **Mod-$k$ extension**: dgMARK naturally extends to mod 3 or higher by partitioning the vocabulary into multiple groups, enabling fine-grained watermark representations. However, as the signal is distributed across more groups, the statistical bias per group may weaken, potentially reducing detectability. Text quality is largely preserved, as no probability reweighting is applied. Empirically, the mod 3 variant improves TPR from 0.540 to 0.782, with only a slight decrease in TNR from 1.000 to 0.995, confirming feasibility with comparable text quality.
>   - | Method         | PPL ↓ | TNR ↑ | TPR ↑ |
> | -------------- | ----- | ----- | ----- |
> | Non-Watermark  | 4.03  |       |       |
> | dgMARK (mod 3) | 4.56  | 0.995 | 0.782 |
> | \+ 3beam       | 4.98  | 0.995 | 0.966 |
>
> - **Multi-token generation**: While our current evaluation focuses on sequential decoding, in principle, dgMARK is compatible with more general strategies. In multi-token generation, the parity constraint can be applied per position or used to prioritize candidate sets that jointly satisfy the watermark condition. Similarly, in beam search, the watermark signal can be incorporated by favoring beams with higher parity-matching scores.
>
> >[Q3] Clarification on argmax vs. sampling in the decoding process
>
> The argmax in Algorithm 2 is used to select the next position to unmask, based on the reward $r_j$, i.e., it determines the index $k^\star$ at each step. Greedy or Multinomial sampling governs token selection at the chosen position: Greedy selects $v_j = \arg\max p_θ(⋅)$, while Multinomial sampling draws $v_j \sim p_θ(⋅)$.
>
> >[Q4] Robustness to misalignment
>
> Our method remains robust to insertion and deletion via two key mechanisms:
> - The parity-matching condition repeats across positions, so a shift flips alignment rather than destroying the watermark signal. The windowed parity matching ratio deviates systematically from the baseline (≈ 0.5), becoming > 0.5 when aligned and < 0.5 when misaligned.
> - Aggregating statistics over local windows produces a characteristic bimodal distribution, clearly distinguishable from the unimodal distribution of non-watermarked text.
>
> As shown in Appendix Figures 5 and 6, this enables reliable detection under insertion, deletion, and token-level misalignment.
>
> [1] Gloaguen et al., "Watermarking Diffusion Language Models" (ICLR 2026)

---

> > ### Author Rebuttal · Reviewer_N4Ns · 2026-04-04
> >
> > Thank you for the detailed rebuttal and clarifications. I have a few suggestions to strengthen the paper:
> >
> > 1. Please provide experiments showing how text quality and watermark detectability vary under different watermark strengths, to better characterize the trade-off.
> >
> > 2. Please include results for settings where multiple tokens are generated per step. Currently, generating a single token per step in dLLM may be slower than standard LLM decoding, while generating multiple tokens per step is a key advantage of dLLM. Empirical validation of this aspect would be helpful.
> >
> > Overall, the idea is largely heuristic, and more comprehensive experiments are needed to compensate for the current lack of theoretical justification.

---

> > > ### Author Response · Authors · 2026-04-06
> > >
> > > We sincerely thank the reviewer for the constructive suggestions. We agree that stronger empirical validation greatly improves the paper. Following your recommendations, we conducted two additional experiments, which will be included in the revision:
> > >
> > > >**1. Trade-off Between Text Quality and Detectability**
> > >
> > > To characterize the trade-off between text quality and detectability under varying watermark strengths, we use the variation of the modulo base $k$ in dgMARK’s token partition scheme as our control mechanism.
> > >
> > > Theoretically, adjusting $k$ directly modulates the watermark strength. Increasing the modulo base $k$ imposes a stricter selective matching condition, which strengthens the embedded watermark signal and enhances detectability. However, this stricter constraint correspondingly restricts the model's generation space, which can theoretically degrade text quality.
> > >
> > > To empirically evaluate this trade-off, we conducted additional experiments extending the partition to mod 3 and mod 4. As shown in the table, our results clearly illustrate this dynamic: as the watermark strength increases via a larger $k$, the stronger selectivity dominates. Detectability improves significantly (TPR increases from 0.540 to 0.945), while incurring only a mild, acceptable cost to text quality (Perplexity increases from 4.44 to 4.84).
> > >
> > >
> > > | Method                 | PPL ↓ | Z = 4 |       |        |       | TPR@FPR  ↑ |        |       |       |
> > > |------------------------|:-----:|:-----:|:-----:|:------:|:-----:|:----------:|:------:|:-----:|:-----:|
> > > |                        |       | **FPR ↓** | **TNR ↑** | **TPR ↑** | **FNR ↓** | **10%** | **1%** | **0.1%** | **0.01%** |
> > > | Non-watermark          |  4.03 |       |       |        |       |            |        |       |       |
> > > | dgMARK                 |  4.44 | 0.000 | 1.000 |  0.540 | 0.460 |    97.86   |  91.98 | 76.47 | 60.96 |
> > > | dgMARK + 3beam         |  4.75 | 0.000 | 1.000 |  0.963 | 0.037 |   100.00   |  99.54 | 98.62 | 97.25 |
> > > | dgMARK (mod 3)         |  4.56 | 0.005 | 0.995 |  0.782 | 0.218 |    98.32   |  95.53 | 91.06 | 82.68 |
> > > | dgMARK (mod 3) + 3beam |  4.98 | 0.005 | 0.995 |  0.966 | 0.034 |   100.00   | 100.00 | 98.52 | 98.03 |
> > > | dgMARK (mod 4)         |  4.84 | 0.000 | 1.000 |  0.945 | 0.055 |    99.31   |  98.62 | 98.62 | 94.48 |
> > > | dgMARK (mod 4) + 3beam |  5.06 | 0.000 | 1.000 |  0.989 | 0.011 |   100.00   | 100.00 | 98.92 | 98.92 |
> > >
> > >
> > >
> > > >**2. Multi-Token Generation Per Step**
> > >
> > > To validate dgMARK's compatibility with parallel decoding, we implemented a multi-token decoding scheme inspired by Fast-dLLM [1] using the LLaDA 1.5 model. The standard Fast-dLLM method accelerates decoding by simultaneously selecting all tokens with a confidence score > 0.9; if no token meets this threshold, it falls back to selecting the single token with the highest confidence. To integrate dgMARK, we adapted this scheme to operate over the watermark's valid partition: our variation selects all "green" tokens with a confidence > 0.9. If no green token exceeds this threshold, we simply select the green token with the highest confidence.
> > > As shown in the table below, dgMARK successfully adapts to this parallelized setting, maintaining comparable text quality (PPL 4.42 vs. non-watermarked 4.01) while retaining strong detectability (TPR 0.502).
> > >
> > >
> > > | Method                      | PPL ↓ | Z = 4 |       |        |       | TPR@FPR  ↑ |       |       |       |
> > > |-----------------------------|:-----:|:-----:|:-----:|:------:|:-----:|:----------:|:-----:|:-----:|:-----:|
> > > |                             |       | **FPR ↓** | **TNR ↑** | **TPR  ↑** | **FNR ↓** |     **10%**    |   **1%**  |  **0.1%** | **0.01%** |
> > > | Non-watermark               |  4.03 |       |       |        |       |            |       |       |       |
> > > | dgMARK                      |  4.44 | 0.000 | 1.000 |  0.540 | 0.460 |    97.86   | 91.98 | 76.47 | 60.96 |
> > > | Non-watermark (fast dLLM) |  4.01 |       |       |        |       |            |       |       |       |
> > > | dgMARK (fast dLLM)        |  4.42 | 0.000 | 1.000 |  0.502 | 0.498 |    96.77   | 88.02 | 77.42 | 57.60 |
> > >
> > >
> > > Despite a slight dilution of the watermark signal from simultaneous token commitments, dgMARK remains highly effective and detectable in this practical, fast-decoding setup. We believe these additions provide the comprehensive empirical validation needed to better characterize the method’s theoretical trade-offs and practical advantages.
> > >
> > > [1] Wu et al., "Fast-dLLM: Training-free Acceleration of Diffusion LLM by Enabling KV Cache and Parallel Decoding" (ICLR 2026)

---

### Official Review · Reviewer_6FPr · 2026-03-11

**Soundness:** 3
**Presentation:** 3
**Significance:** 3
**Originality:** 3
**Overall Recommendation:** 4
**Confidence:** 3

**Summary:**

This paper proposes dgMARK, a watermarking method for discrete diffusion language models (dLLMs). The key motivation is that dLLMs can iteratively denoise masked tokens in an arbitrary order, making conventional watermarking methods—designed for left-to-right autoregressive decoding and relying on probability reweighting—less suitable. dgMARK instead uses decoding-order control as the watermarking channel: it applies a binary hash over candidate tokens and enforces a parity-matching constraint to prioritize which positions are filled first, thereby improving watermark detectability without modifying the model’s learned probability distribution. In addition, the paper introduces dgMARK with One-step Lookahead Beam Search to strengthen the watermark signal, and employs a sliding-window detector to improve robustness against post-editing and rewriting.

**Compliance With Llm Reviewing Policy:**

Affirmed.

**Final Justification:**

The author replied to my question. Considering the overall quality of the paper, I maintained my positive score.

**Key Questions For Authors:**

1. Can the authors provide additional experiments on watermark imperceptibility, as well as a security proof (or formal guarantees) for the proposed scheme?
2. If an attacker can collect a large number of watermarked texts, is it possible to infer the secret key? Can the authors include experiments evaluating key-recovery or key-inference attacks under this setting?
3. When the attacker knows the watermarking algorithm details, does there exist an efficient watermark-removal attack that uses only a small number of edits to push the z-score or (z_{\text{win}}) below the detection threshold?
4. Beyond the standard decoding-order heuristics considered in the paper, are there other decoding mechanisms for diffusion language models, and is dgMARK compatible with them? When using a watermark-intervened decoding order, can the model retain its original performance under these alternative decoding schemes?

**Limitations:**

The discussion of limitations and potential negative societal impacts is insufficient. The authors are encouraged to explicitly add and discuss: (1) formal justification and dedicated experiments on the imperceptibility of the watermarked text, rather than relying solely on PPL and downstream task metrics; (2) security and robustness evaluations under stronger adversary settings.

**Strengths And Weaknesses:**

**Strengths:**

1. The method is novel: it leverages the non-sequential decoding property of diffusion language models as the key angle for watermark embedding and detection. The proposed variant further improves watermark detectability, and the window-based detector enhances robustness.
2. The approach is plug-and-play, requires no additional training, and is compatible with standard decoding strategies used in diffusion language models.
3. The paper provides a systematic evaluation of watermarked text quality, including perplexity (PPL) and downstream task performance, and also evaluates robustness under multiple attacks. The watermarking method performs well under the experimental settings considered in the paper.

**Weaknesses:**

1. Although the experiments evaluate text quality and downstream capabilities, the paper does not include a dedicated imperceptibility study for the watermark.
2. Under a strong adversary who knows the watermarking algorithm details but not the secret key, the paper lacks a deeper security analysis and a systematic evaluation.

---

> ### Author Rebuttal · Authors · 2026-03-30
>
> We appreciate the reviewer’s constructive comments and positive feedback. We address the specific questions below.
>
> ---
> >**[W1, Q1]** Additional experiments on watermark imperceptibility and formal security guarantees.
>
> Regarding watermark imperceptibility, our manuscript includes quantitative quality metrics and full, representative text examples in the Appendix (Figure 16, Tables 12–14) demonstrating preserved fluency. We agree that direct imperceptibility evaluation is important and will conduct a GPT-as-a-judge evaluation to compare the quality of watermarked and non-watermarked text as a proxy for imperceptibility. A brief summary is provided below, where the evaluation prompt is adapted from [1].
>
> | **Method**        | **Coherence↑** | **Clarity↑** | **Naturalness↑** | **Overall↑** |
> |:---------------|:----------:|:--------:|:------------:|:--------:|
> | Non-Watermark | 4.63      | 4.65    | 4.62        | **4.63**    |
> | dgMARK        | 4.65      | 4.64    | 4.61        | **4.63**    |
> | + 3beam       | 4.61      | 4.62    | 4.61        | 4.61    |
> | + 5beam       | 4.58      | 4.58    | 4.57        | 4.57    |
> | + 8beam       | 4.59      | 4.60    | 4.56        | 4.58    |
>
> Regarding formal security guarantees, we respectfully note that establishing rigorous cryptographic bounds for LLM watermarks remains a significant open challenge. Recent studies [2] demonstrate that even state-of-the-art schemes previously assumed to be secure are vulnerable to watermark stealing and spoofing attacks. Since natural language distributions are complex and adversaries can approximate rules via API queries, absolute guarantees are challenging.
>
> Similar to other recent advancements that prioritize preserving text quality [1], our primary contribution is a practical scheme that achieves robust empirical detectability without explicitly reweighting token probabilities. We will expand the discussion to clarify these limitations and highlight formal guarantees as future work.
>
> > **[W2, Q2]** Security analysis under a strong adversary and key inference.
>
> As noted in our previous response, providing strict formal security guarantees against a well-resourced adversary remains an open challenge in LLM watermarking. While a strong adversary could potentially observe position-dependent statistical patterns, exploiting this is practically challenging for two main reasons:
> - To successfully infer the secret key or precise token-to-group mapping, an adversary must collect an exceptionally large volume of watermarked text. Because the watermark relies on aggregated statistical bias, exact key recovery from frequency information alone is highly non-trivial.
> - To further secure the scheme, we can alter the hashing function at regular intervals (e.g., every 16 tokens). This dynamic key rotation severely limits the stationary statistical data available to an attacker, increasing the complexity of inference attacks.
>
> We will expand our security analysis in the revised manuscript to discuss these practical mitigations and inherent challenges of formal security proofs.
>
> > **[Q3]** Watermark-removal attacks with minimal edits.
>
> Evading detection with few targeted edits is highly challenging, as our detection relies on a global statistical bias. To evade detection, an attacker must push the parity-matching ratio down to $\approx 0.5$ (the expected baseline for unwatermarked text). Achieving this requires a significant number of edits distributed across the entire sequence. Localized modifications are mathematically insufficient to weaken this aggregated signal, and forcing widespread edits inevitably degrades text quality.
>
> > **[Q4]** Compatibility with other dLLM decoding methods.
>
> dgMARK is fundamentally compatible with any decoding strategy for dLLMs, as it can simply be applied on top of existing mechanisms.
>
> To demonstrate this, we evaluated dgMARK under decoding strategies beyond the standard heuristics in the main text. We applied dgMARK to entropy- and margin-based decoding by incorporating our watermark constraint into their position selection processes. As detailed in Appendix Tables 9 and 10, the model retains its original performance well under these alternative schemes, with reliable detectability and only modest perplexity changes (e.g., PPL shifts from 4.19 to 4.51 for entropy-based, and from 3.95 to 4.40 for margin-based decoding).
>
> These results confirm that our method acts as a flexible, compatible layer integrable with various decoding mechanisms without significantly degrading generation quality.
>
> > **[L1]** Limitations and societal impacts.
>
> We appreciate the suggestion and will revise the manuscript to provide a clearer discussion of the limitations and potential societal impacts, incorporating the points outlined above.
>
> [1] Jiang et al., "MirrorMark: A Distortion-Free Multi-Bit Watermark for Large Language Models" (ArXiv 2026)
> [2] Jovanović et al., "Watermark Stealing in Large Language Models" (ICML 2024)

---

> > ### Author Rebuttal · Reviewer_6FPr · 2026-04-03
> >
> > Thank you for the response. After carefully reviewing all feedback, I have decided to maintain my original score.

---

### Official Review · Reviewer_iFBj · 2026-03-12

**Soundness:** 3
**Presentation:** 3
**Significance:** 3
**Originality:** 2
**Overall Recommendation:** 4
**Confidence:** 3

**Summary:**

This paper proposed dgMARK, a decoding-guided watermarking method for discrete diffusion language models. Specifically, dgMARK steers the decoding order toward positions where the sampled tokens satisfy a simple parity constraint induced by a binary hash, without explicitly reweighting the probabilities. By extensive evaluations across different models and on different tasks, dgMARK shows less generation quality degradation while providing comparable or even better detectability, outperforming the previous watermarking baselines. Besides, dgMARK is also robust against post-editing operations, including insertion, deletion, substitution, and paraphrasing.

**Compliance With Llm Reviewing Policy:**

Affirmed.

**Key Questions For Authors:**

1. What is $i$ in Algorithm 1 and 2? The decoding step?

1. What is the hashing function used in this paper?


1. In Algorithm 2, does every position have a different parity-matching set, or is the parity-matching set the same for all positions? According to the definition of the parity-matching set, all the odd positions should share the same set, while all the even positions should share another set. If I understand it correctly, we have $G_j=R_{j+1}, \forall j$, i.e., the green and red lists are switched at every next position. If so, what does the $G_j$ in the Require line mean?

1. In my opinion, dgMARK also implicitly reweights the token distribution by rejection sampling. In other words, positions with tokens not in the green list are rejected, remasked, and resampled at future steps, which gives the green list indices more opportunities and is equivalent to making the token distribution more biased towards green list indices. As a result, we can expect to see more significant reweighting under multinomial sampling, and greedy sampling is more resistant to this implicit reweighting. In fact, we can observe less performance drop under greedy decoding in Table 3, and a non-negligible detectability drop under greedy decoding in Table 4, which further supports my argument. Could the authors provide a further discussion on this point?

1. Although dgMARK brings less generation quality degradation compared with other watermarking baselines and almost keeps the same accuracy as the non-watermarked baseline on MMLU, we can still observe a non-negligible performance drop on GSM8K, and a more significant drop on HumanEval. From my perspective, the math reasoning and code generation tasks are more sensitive to decoding order as they require more complicated reasoning, and controlling the decoding order is very likely to degrade the reasoning ability of diffusion LLMs. Could the authors provide a further discussion on this point?

1. Could the authors provide a more detailed description on Figure 5 and 6? What is the x-axis?

1. What is the definition of parity-matching ratio?

**Limitations:**

1. As mentioned in **Weaknesses** and **Questions**, on tasks that require more complicated reasoning, dgMARK still brings non-negligible performance degradation. Overcoming this challenge is still a critical problem.

1. The claim of no probability reweighting is not rigorous. In my opinion, although without explicit probability reweighting, dgMARK still implicitly reweights the token distribution by rejection sampling.

**Strengths And Weaknesses:**

### Strengths
1. The structure and presentation of this paper are clear.
1. The performance of the proposed method is good. It outperforms the previous baselines by less generation quality degradation while providing comparable or even better detectability.
1. The problem setup is clear, and the algorithm is intuitive and reasonable to me.
1. The evaluations are extensive and solid.

### Weaknesses
1. Some denotations are confusing. Please see **Questions**.
1. Although the proposed method outperforms the previous baselines, it still brings a non-negligible performance drop on the math reasoning and code generation tasks.
1. Some figures do not provide informative descriptions. Please see **Questions**.
1. The claim of no probability reweighting is not rigorous enrough to me. Please see **Questions**.

---

> ### Author Rebuttal · Authors · 2026-03-30
>
> We appreciate the reviewer’s valuable insights and acknowledgment of the strengths of our approach. We address your specific questions below.
>
> ---
> >**[W1, Q1, Q3]** Clarification of notation in Algorithms 1, 2 ($i$, $\mathcal{G}_j$)
>
> We thank the reviewer for their careful reading. Your understanding of the mechanics is entirely correct.
>
> To clarify your specific questions:
> - $i$ denotes the current decoding step.
> - The parity-matching sets simply alternate rather than being uniquely generated for every position. At every odd position, $\mathcal{G}_i$ contains tokens with a hash value of 1. At every even position, $\mathcal{G}_i$ contains tokens with a hash value of 0.
>
> As the reviewer noted, we do not need to explicitly specify each $\mathcal{G}_j$ in the Require line of Algorithm 2. Defining the hashing function is sufficient, and the alternating $\mathcal{G}_j$ sets follow as a direct consequence. We will update the notations in the revised manuscript to make these definitions explicit.
>
> >**[Q2]** Hashing function used in the method
>
> In principle, any hash function yields similar results provided that it is balanced. For simplicity in our implementation, we utilized a basic parity function applied to the token ID. We will explicitly state this detail in the revised manuscript.
>
> >**[W2, Q5]** Performance drop on GSM8K and HumanEval
>
> A fundamental trade-off between watermark detectability and generation quality is unavoidable and well-documented, even in standard Autoregressive LLMs [1].
>
> Regarding the performance drops on GSM8K and HumanEval, we respectfully clarify that this is not due to a specific vulnerability to decoding order. Rather, it stems from the deterministic nature of these low-entropy tasks, which carry strict syntactic and logical constraints. Forcing token choices to satisfy a watermark naturally degrades performance in such rigid contexts. It is a known, universal challenge when watermarking code and math across all LLM architectures [2]. To further support this claim, we provide a comparison with a concurrent dLLM watermarking method [3] in the table below.
>
> | **Method**         | **MMLU↑** | **GSM8K↑** | **HumanEval ↑** |
> | :-------------- | :-----: | :------: | :-----------: |
> | [3] ($δ = 1$)    | 0.586 | 0.733  | **0.232**       |
> | [3] ($δ = 2$)    | 0.535 | 0.644  | 0.128       |
> | dgMARK + 3beam | **0.649** | **0.774**  | 0.207       |
>
> The results highlight the inherent difficulty of these benchmarks for watermarking, while our method remains competitive. We will expand this discussion and include these references in the revised manuscript.
>
> >**[W3, Q6, Q7]** Clarification of Figures 5, 6 and the parity-matching ratio
>
> We thank the reviewer for highlighting this omission. These points are closely related, as the x-axis in Figures 5 and 6 represents the parity-matching ratio computed over a sliding window.
>
> The parity-matching ratio is the proportion of tokens in a sequence of length $n$ that satisfy the parity condition (i.e., whether the token $y_i$ belongs to the parity-matching set $\mathcal{G}_i$). It is formally defined as:
>
> $\frac{1}{n} \sum_{i=1}^{n} \mathbf{1}[y_i \in \mathcal{G}_i]$
>
> In watermarked text, this ratio naturally tends to exceed 0.5. We will update the main text and figure captions to make these definitions explicit.
>
> >**[W4, Q4]** Concern that rejection sampling in dgMARK implicitly biases the token distribution
>
> We thank the reviewer for the detailed analysis. First, we wish to emphasize that our core claim "strictly no probability alteration" is mathematically exact for greedy decoding. Because greedy decoding deterministically selects the $\arg\max$ token without resampling, it is entirely immune to implicit bias and perfectly preserves the original output path.
>
> Regarding the multinomial setting, the reviewer is correct. The rejection step inadvertently acts as implicit resampling, biasing the distribution toward the matching list. We will clarify this distinction in the text, framing the multinomial step as a practical trade-off. Unlike standard watermarks that distort logits with fixed scalars, our implicit rejection acts as "soft resampling" that preserves the relative probabilities among accepted tokens, trading a bounded KL-divergence for a gain in detectability.
>
> To transparently demonstrate this Pareto frontier, we have implemented a strictly unbiased multinomial baseline (sampling index prior to rejection), which preserves the exact unconditioned distribution but reduces detectability, as expected. We will update our manuscript accordingly to reflect these clarified boundaries and include the new ablation results.
>
> >**Regarding [L1, 2]**, we will update the limitations section accordingly in the final version.
>
> [1] Kirchenbauer et al., "A Watermark for Large Language Models" (ICML 2023)
> [2] Lee et al., "Who Wrote this Code? Watermarking for Code Generation" (ACL 2024)
> [3] Gloaguen et al., "Watermarking Diffusion Language Models" (ICLR 2026)

---

> > ### Author Rebuttal · Reviewer_iFBj · 2026-04-02
> >
> > Thank the authors for the response. I think it is necessary to clarify the claim of no probability reweighting in the future version. I will maintain my positive rating.

---

### Official Review · Reviewer_ovQc · 2026-03-13

**Soundness:** 3
**Presentation:** 2
**Significance:** 3
**Originality:** 3
**Overall Recommendation:** 4
**Confidence:** 3

**Summary:**

This paper proposes dgMARK, a novel decoding-guided watermarking strategy specifically designed for discrete diffusion language models. Unlike traditional autoregressive watermarking methods that explicitly bias token probabilities using predefined lists, dgMARK capitalizes on the order-agnostic generation capabilities of dLLMs. It embeds a watermark by prioritizing the unmasking order of positions where high-reward candidate tokens satisfy a simple parity constraint determined by a binary hash. The authors introduce a lookahead beam search variant to enhance signal strength without modifying learned probabilities, and they utilize a sliding-window z-statistic detector to ensure robustness against post-editing.

**Compliance With Llm Reviewing Policy:**

Affirmed.

**Final Justification:**

The authors have addressed my concern in their rebuttal.

**Key Questions For Authors:**

See weaknesses

**Limitations:**

Yes

**Strengths And Weaknesses:**

**Strengths**

- The approach exploits the decoding order flexibility of diffusion language models, offering a novel, structural alternative to standard probability-biasing paradigms.

- dgMARK operates seamlessly as a wrapper and is compatible with a wide range of common dLLM decoding strategies.

- The sliding-window detection mechanism provides strong empirical resilience against token-level perturbations, maintaining reliable detection under random deletions, insertions, and substitutions



**Weaknesses**


- The empirical evaluation would be significantly strengthened by including comparisons against more recent watermarking baselines specifically designed for discrete diffusion language models.

- The authors claim that because they do "not explicitly reweighting the model's learned probabilities," the method preserves quality. However, by altering the decoding order $\pi$ dynamically based on whether candidate tokens match the parity constraint $G_i$, $\pi$ becomes dependent on the watermark key $\xi$. Even if the token-level probability is unadjusted, changing the sequence of conditioning variables fundamentally alters the joint distribution of the final generated sequence. There are no formal proofs or bounds provided to prove that this decoding-guided shift introduces zero or bounded distortion to the global distribution.

- The proposed one-step lookahead beam search, which is necessary to achieve the strongest detectability, introduces substantial latency. For instance, utilizing a beam size of 5 increases the generation overhead by 3.8x compared to the non-watermarked baseline.

---

> ### Author Rebuttal · Authors · 2026-03-30
>
> We appreciate the reviewer’s careful review and recognition of the novelty of our approach. We address the reviewer’s concerns below.
>
> -----
> > **[W1]** Suggestion to include recent dLLM watermarking baselines.
>
> We thank the reviewer for this constructive suggestion. While some recent baselines (including [1]) are highly relevant, we respectfully note that it is concurrent work. For our initial submission, we ensured a rigorous evaluation by establishing a strong, reasonable baseline using PATTERN-MARK.
>
> Nevertheless, we completely agree that comparing against this new dLLM-specific method significantly strengthens our empirical evaluation. As requested, we have conducted the additional experiments using the baseline [1].
>
> | **Method**         | **PPL ↓** | **FPR ↓** | **TNR ↑** | **TPR ↑** | **FNR ↓** |
> | :-------------- | :-----: | :-----: | :-----: | :-----: | :-----: |
> | [1] ($δ = 1$)    | 4.87  | 0.000 | 1.000 | 0.484 | 0.516 |
> | [1] ($δ = 2$)    | 6.04  | 0.000 | 1.000 | 0.917 | 0.083 |
> | dgMARK         | **4.44**  | 0.000 | 1.000 | 0.540 | 0.460 |
> | dgMARK + 3beam | 4.75  | 0.000 | 1.000 | **0.963** | **0.037** |
>
>
> The new results demonstrate that our method achieves competitive watermark detectability while preserving text quality, consistent with our main claims. We will gladly incorporate these new experiments and discussions into the revised manuscript.
>
> >**[W2]** Concern that decoding-order changes may alter the joint distribution without guarantees.
>
> We thank the reviewer for the insightful comment. We completely agree that dynamically altering the decoding order $\pi$ based on the watermark key $\xi$ affects the joint distribution of the final generated sequence.
>
> As recently discussed in [2], while the chain rule of probability dictates that any decoding order should theoretically yield the exact same joint distribution, they differ in practice. This discrepancy arises from training imperfections and the model's approximated conditional distributions. Providing formal guarantees or bounds on this distortion remains an open theoretical challenge in the field of any-order generation, similar to the current lack of formal proofs explaining exactly why confidence-based decoding outperforms other arbitrary orders.
>
> Our claim in the paper regarding preserved quality specifically highlights that our method does not explicitly modify or reweight the model’s learned token probabilities (e.g., logits). In prior watermarking approaches, directly biasing token selection requires manual alterations and introduces additional hyperparameter choices to balance quality and watermark strength. By avoiding these manual alterations, our method reduces the burden of hyperparameter tuning.
>
> Ultimately, our scheme naturally aligns with the spirit of dLLM, which is designed to allow for any-order decoding. By leveraging this inherent flexibility, we achieve watermarking through context conditioning rather than probability biasing. While we acknowledge the reviewer's point that this indirectly influences the global distribution, we empirically observe that the impact on text quality is minimal, as demonstrated in our experimental results.
>
> > **[W3]** Concern that beam search introduces significant latency.
>
>
> We thank the reviewer for this valuable comment. We would like to clarify that the proposed one-step lookahead beam search is an optional component designed to further enhance detectability, rather than a required part of our method.
>
> In our evaluation, even under the default decoding setting (i.e., without beam search), our method already achieves meaningful and competitive detectability. For example, compared to the newly introduced baseline [1] with $\delta = 1$, which achieves a TPR of 0.484 at PPL 4.87, whereas our method achieves a higher TPR of 0.540 at a lower PPL of 4.44, indicating a more favorable detectability–quality trade-off.
>
> Furthermore, we observe that applying a small beam size (e.g., $k=3$) substantially strengthens detectability (e.g., TPR up to 0.963 with modest PPL increase), indicating that beam search acts as an effective enhancement rather than a necessity.
>
> We agree that this design introduces a trade-off between computational cost and detection performance. Beam search can therefore be selectively applied in scenarios where higher detection reliability is required, while the default decoding setting remains suitable for latency-sensitive applications with only limited impact on performance.
>
> Overall, our method provides flexibility in balancing detection strength and generation efficiency depending on deployment requirements. We will revise the manuscript to clarify this distinction and highlight the optional role of beam search more explicitly.
>
> [1] Gloaguen et al., "Watermarking Diffusion Language Models" (ICLR 2026)
> [2] Kim et al., "Train for the Worst, Plan for the Best: Understanding Token Ordering in Masked Diffusions" (ICML 2025)

---

> > ### Author Rebuttal · Reviewer_ovQc · 2026-04-01
> >
> > Thank you for your detailed rebuttal. I will keep my positive rating.

---

### Decision · Program_Chairs · 2026-04-30

**Decision:**

Accept (regular)

**Comment:**

This paper introduces dgMARK, a watermarking framework for discrete diffusion language models that exploits their sensitivity to token unmasking order. Rather than altering token probabilities directly, the method steers the decoding order toward positions whose candidate tokens satisfy a hash-based parity rule, making it a lightweight and decoding level watermarking strategy. The approach is designed to integrate with standard dLLM decoding policies and includes a sliding window detector to improve robustness under post-editing and paraphrasing. The method is technically sound and clean in presentation.


The most common weakness is the lack of theoretical guarantees. Several reviewers felt the method is largely heuristic at present, with insufficient formal analysis of detectability, robustness, and especially distributional distortion.  Either formal guarantees or at least bounds on how much distortion this introduces would make the paper stronger than the current form. Moreover, the baseline coverage is not comprehensive, the watermarks still impose a non-negligible quality cost, especially on harder tasks such as math reasoning and code generation, and the strongest version of the method incurs substantial latency overhead while still causing noticeable degradation on some reasoning and code tasks

After rebuttal most of the points are addressed. Although the paper still hardly has theoretical supports, the current form looks good to all reviewers. It is encouraged to push the theoretical anlysis further or provide more empirical justification to support the paper.